# The Glycosaminoglycan Side Chains and Modular Core Proteins of Heparan Sulphate Proteoglycans and the Varied Ways They Provide Tissue Protection by Regulating Physiological Processes and Cellular Behaviour

**DOI:** 10.3390/ijms241814101

**Published:** 2023-09-14

**Authors:** Brooke L. Farrugia, James Melrose

**Affiliations:** 1Department of Biomedical Engineering, Faculty of Engineering and Information Technology, University of Melbourne, Melbourne, VIC 3010, Australia; brooke.farrugia@unimelb.edu.au; 2Graduate School of Biomedical Engineering, University of New South Wales, Sydney, NSW 2052, Australia; 3Raymond Purves Laboratory of Bone and Joint Research, Kolling Institute of Medical Research, Northern Sydney Local Health District, Royal North Shore Hospital, St. Leonards, NSW 2065, Australia; 4Sydney Medical School (Northern), University of Sydney at Royal North Shore Hospital, St. Leonards, NSW 2065, Australia

**Keywords:** HS, cellular regulation, tissue development, ECM remodeling, tissue protection, tissue homeostasis, neural plasticity, neurotransduction, phototransduction, neurogenesis, regulation of angiogenesis, matrix stabilization, cellular proliferation and differentiation

## Abstract

This review examines the roles of HS–proteoglycans (HS–PGs) in general, and, in particular, perlecan and syndecan as representative examples and their interactive ligands, which regulate physiological processes and cellular behavior in health and disease. HS–PGs are essential for the functional properties of tissues both in development and in the extracellular matrix (ECM) remodeling that occurs in response to trauma or disease. HS–PGs interact with a biodiverse range of chemokines, chemokine receptors, protease inhibitors, and growth factors in immune regulation, inflammation, ECM stabilization, and tissue protection. Some cell regulatory proteoglycan receptors are dually modified hybrid HS/CS proteoglycans (betaglycan, CD47). Neurexins provide synaptic stabilization, plasticity, and specificity of interaction, promoting neurotransduction, neurogenesis, and differentiation. Ternary complexes of glypican-1 and Robbo–Slit neuroregulatory proteins direct axonogenesis and neural network formation. Specific neurexin–neuroligin complexes stabilize synaptic interactions and neural activity. Disruption in these interactions leads to neurological deficits in disorders of functional cognitive decline. Interactions with HS–PGs also promote or inhibit tumor development. Thus, HS–PGs have complex and diverse regulatory roles in the physiological processes that regulate cellular behavior and the functional properties of normal and pathological tissues. Specialized HS–PGs, such as the neurexins, pikachurin, and Eyes-shut, provide synaptic stabilization and specificity of neural transduction and also stabilize the axenome primary cilium of phototoreceptors and ribbon synapse interactions with bipolar neurons of retinal neural networks, which are essential in ocular vision. Pikachurin and Eyes–Shut interactions with an α-dystroglycan stabilize the photoreceptor synapse. Novel regulatory roles for HS–PGs controlling cell behavior and tissue function are expected to continue to be uncovered in this fascinating class of proteoglycan.

## 1. Introduction

Heparan sulphate proteoglycans (HS–PGs) are a diverse group of highly functional proteins (Figure 1, Figure 2 and Figure 3) that are widely distributed in mammalian tissues and have well-known roles in tissue development and repair processes [1,2,3,4,5,6,7,8,9,10,11,12,13,14,15,16,17]. HS–PGs have a diverse range of interactive ligands, numbering in excess of 400 HS-binding protein members [18,19,20,21]; a murine acute pancreatitis model has expanded this to 786 HS-binding members [18,22,23]. Synaptic neurexin–ligand interactions also number in several hundred combinations that provide synaptic connectiveness, the specificity of neuronal action, and synaptic plasticity.

HS glycosaminoglycans (GAG) evolved over a 500-million-year period under strict evolutionary selection criteria as a regulatory molecule in the glycocalyx, which displayed molecular recognition and information-storage properties capable of controlling cellular behavior and a wide range of essential physiological life processes. HS side chains are important function-defining components of HS–PGs, which convey a diverse range of cellular and physiological regulatory functions. Proteins that interact with HS include a range of growth factors, neurotrophins, cytokines, chemokines, morphogens, extracellular matrix (ECM) structural proteins, cell-adhesion molecules, proteases, and protease inhibitory proteins [24]. In a comprehensive review of proteoglycan nomenclature conducted in 2015 [25], a total of 22 HS–PGs were identified. Additional HS–PGs have since been identified, including the neurexin α, β, γ family with roles in synapse stabilization, synaptogenesis, synaptic plasticity, and neural function [26,27,28,29]. Further, two retinal basement membrane HS-PGs, Eyes–shut [30] and Pikachurin [31,32], have also been identified with roles in the stabilization of the rod and cone photoreceptor axenome primary cilium, inter photoreceptor ECM, and the photoreceptor ribbon synapses, which interact with bipolar neurons facilitating photo-transduction and neurotransmission in retinal neural networks essential for high-quality ocular vision. A few established PGs have also now been shown to contain HS side chains in specialized tissue contexts. These include a form of aggrecan found in bovine rib growth plate cartilage [33], a specialized form of CD44 called Epican, and the HS/CS dually modified co-receptors CD47 [34] and betaglycan [35]. CD47, first recognized as a 50 kDa protein that co-purified with αvβ3 integrin in the placenta and neutrophil granulocytes, also occurs as a 250 kDa PG-bearing CS and HS GAG chains [36,37]. The GAG chains of CD47 are crucial in the inhibition of T cell receptor signaling following the ligation of CD47 by thrombospondin-1 [34]. The GAG-modified form of NRP-1 regulates VEGFR2 protein expression [35,38] and modulates VEGF signaling, providing novel insights into physiological and pathological angiogenic processes. NRP-1 is modified with either HS or CS but does not contain both GAGs on the same molecule [35]. Betaglycan is a dually modified CS-HS 250–280 kDa transmembrane PG co-receptor for the TGF-β superfamily forming a multi-functional [39,40] homodimer at the cell surface containing inhibin, FGF-2, Wnt, and TGF-β-binding sites [41,42,43,44,45,46,47,48,49]. Dual modified CS/HS cell surface PG receptors, such as betaglycan and CD47, play crucial roles in the regulation of key developmental-signaling events involving the Wnt, Hh, TGF-β, and FGF cell-signaling pathways [50]. When substituted with HS, Wnt signaling by CD47 and betaglycan is inhibited while CS promotes Wnt signaling. CS also provides inhibitory cues over neural development, while HS has an opposing effect mediated through the LAR RPTP-σ PG expressed by neurons [51,52]. Glypican HS–PGs also regulate round-about receptor interactions with Slit 1–3 and netrin neurotrophic factors that direct axonal development controlling neural development and network formation [53,54,55].

## 2. The Diversity of Cell Surface, Extracellular, and Cytoplasmic HS–PGs

### 2.1. Cytoplasmic HS–PGs

Serglycin is a small molecular weight (17.6 kDa core protein) intracellular heparin-proteoglycan present in cytotoxic T lymphocytes, leucocytes, and NK cells found in blood. Serglycin is also expressed in glioma, where the cross-talk of activated astrocytes with glioma cells enhances serglycin production. This is a predictive biomarker of poor survival [56,57]. Serglycin can also be secreted and incorporated into the ECM and displays highly divergent glycosylation patterns in different cell types. The serglycin core protein has a central 16 amino acid region substituted with long GAG chains. Serglycin is expressed by cells of haematopoietic origin, including neutrophils, lymphocytes, monocytes, macrophages, platelets, megakaryocytes, and mast cells [16,58,59,60,61], as well as endothelial and embryonic stem cells [17,60]. Intervertebral disc cells and chondrocytes also express serglycin [62,63,64], with IL-1β or TNF-α increasing serglycin expression in vitro. Serglycin levels are also elevated under inflammatory conditions during cartilage and IVD degeneration [62,63,64]. Heparin exclusively substitutes serglycin in connective tissue mast cells [65]. However, mucosal mast cells, activated monocytes, and macrophages contain serglycin substituted with highly sulphated chondroitin-4,6-disulfate; quiescent monocytes contain serglycin substituted with chondroitin-4-sulfate [16].

Serglycin has important functional roles in the formation of storage granules, which contain a range of bioactive molecules whose actions need to be controlled during storage [16]. In mast cells, these compounds include histamine, chymase, tryptase, and carboxypeptidase. Elastase is stored in such granules in neutrophils, cytotoxic T cells store granzyme B, endothelial cells store tissue-type plasminogen activator, while macrophages store TNFα [66]. Serglycin has important roles in the retention of key inflammatory mediators in an inactive form inside storage granules and secretory vesicles, which are released as active components when de-granulation occurs [16]. The release of these components at specific tissue sites can significantly influence the inflammatory process and innate immunity [67,68].

### 2.2. Cell-Associated HS–PGs

#### 2.2.1. The Glypican Family

The glypicans (GPC1-6) are a family of glycosylphosphatidylinositol (GPI)-anchored cell surface HS–PGs that have roles in developmental morphogenesis [69], Wnt, and Hedgehog cell-signaling pathways [69,70,71,72], and they also regulate FGFs and BMPs [4]. All of the glypicans have core proteins ranging between 60–70 kDa in size and contain 2–5 HS chains, athough GPC5 may also have a CS chain in certain tissue contexts. Wnt binds IdoA2S and GlcNS6S motifs in the 3–O sulphated HS chains of GPC3 [72]. GPCs are abnormally expressed in liver and ovarian cancer, mesothelioma, pancreatic cancer, glioma, breast cancer, and neuroblastoma, with most research focusing on the roles of GPC1 and GPC3 [69]. Tumor GPC1 may sequester growth factors and promote tumor cell growth [69,73,74]. GPC2 is highly expressed in ~50% of neuroblastoma cases and is a potential therapeutic target [75,76]. GPC3 is expressed in embryonic development but has a low expression in tissues in maturity and may be of prognostic value in tumours [6].

#### 2.2.2. The Syndecan Family

The syndecans constitute a Type I transmembrane co-receptor PG family consisting of four members, with core proteins ranging in size from 22–43 kDa substituted with 3–5 HS and 1–2 CS side chains [77,78,79]. Cell signalling by syndecan can be initiated in the cytoplasmic tails of the PG [80,81], the core protein, or by GAG side chain interactions [77,78,79]. The syndecan HS chains bind heparin-binding growth factors, such as the FGFs, VEGFs, TGF-β, PDGFs, and ECM structural proteins, including endothelial fibronectin and AT in the endothelium [77,78,79]. The syndecan CS chains may cooperate with HS chains in the binding of midkine, pleiotrophin, and FGFs [82].

### 2.3. Cell Surface Hybrid HS/CS Proteoglycan Co-Receptors

#### 2.3.1. Betaglycan

Betaglycan, a 250–280 kDa multifunctional transmembrane HS/CS co-receptor member of the TGF-β superfamily [39,40], forms a functional homodimer at the cell surface that binds inhibin, activin, FGF-2, Wnt, and TGF [41,42,43,44,45,46,47,48,49]. HS chains of betaglycan bind FGF-2 and Wnt binds specific HS sequences independently of TGF-β binding [83]. Betaglycan binds TGF-beta via its core protein. Thus, betaglycan can bind several classes of growth factors through separate domains [84]. The HS/CS dual modification of betaglycan modulates key cell-signaling pathways, including Wnt, TGF-β, and FGF signaling [83]. Hyperactive Wnt/β-catenin signaling is linked to cancer progression and developmental abnormalities. Betaglycan controls Wnt3a bioavailability, independent of TGF-β co-receptor activity. HS and CS have opposing effects in TβRIII and HS inhibits Wnt signaling, whereas CS promotes Wnt3a signaling and is a key regulatory feature of betaglycan and may be considered a molecular switch over cellular behaviour during tissue development [85]. Betaglycan fragments released from the cell surface by plasmin and MMPs, which act as sheddases, are circulating antagonists to the action of cell-bound betaglycan.

#### 2.3.2. Neuropilin

Neuropilin occurs as two multifunctional Class III semaphorin Type 1 single-pass transmembrane 130–140 kDa HS or CS-substituted glycoprotein receptors (NRP-1, NRP-2). The NRPs bind semaphoring, and VEGFA guides the development of axonal embryonic neural networks. The NRPs are pro-angiogenic in mature tissues, stabilizing VEGF/VEGFR interactions promoting tissue development, repair, and tumor growth [35]. The NRPs are substituted at a single site with either HS or CS but are not dually modified with each GAG.

#### 2.3.3. CD47

CD47 is a multifunctional hybrid dually modified vascular HS/CS transmembrane receptor of hematopoietic cells. The HS substitution on CD47 inhibits T cell receptor signaling by TSP-1 and regulates cellular migration, proliferation, and vascular cell survival in innate and adaptive immune regulation [86,87]. TSP-1 acts via CD47 to inhibit vascular NO signaling and regulates blood pressure [88]. CD47 is a ligand for SIRPα (signal regulatory protein α, also known as SH2-domain-bearing protein tyrosine phosphatase (SHP) substrate-1 (SHPS-1), a brain Ig-like molecule with tyrosine-based activation motifs, or CD172a. These constitute the CD47-SIRPα cell–cell communication system [89,90,91].

CD47 is a widely expressed integrin-associated transmembrane protein encoded by the CD47 gene and is a member of the immunoglobulin superfamily that binds to integrins, thrombospondin-1, and signal-regulatory protein alpha (SIRPα), a protein expressed by macrophages and dendritic cells [92]. Upon binding to CD47, SIRPα initiates a signaling cascade that results in the inhibition of phagocytosis [91]. Phosphorylation of the immunoreceptor tyrosine-based inhibition motifs present on the cytoplasmic tail of SIRPα initiates this response [87]. Subsequent binding and activation of SHP-1 and SHP-2 [src homology-2 (SH2) domain-containing protein tyrosine phosphatases] blocks phagocytosis by preventing the accumulation of myosin-IIA at the phagocytic synapse [88,93,94,95]. CD47 is a critical regulator of innate immune surveillance, while the HS chain of CD47 is necessary for the inhibition of T cell receptor signaling by thrombospondin-1 [34]. CD47 has important roles in the regulation of cellular migration, proliferation, and survival of vascular cells in innate and adaptive immune regulation [86,87]. TSP-1 acts via CD47 to inhibit NO signaling in the vascular system and supports blood pressure regulation by limiting eNOS activation and endothelial-dependent vasorelaxation [88]. CD47-SIRPα interactions represent a cell–cell communication system [89,90,91] termed as an innate immune checkpoint sending a “don’t eat me” signal to macrophages [94]. This allows tumor cells to avoid detection and be cleared by the immune system. A blockade of the CD47–SIRPα interaction using humanized antibodies to CD47 (Hu5F9-G4) has yielded promising results in a number of human malignancies, including pediatric brain tumors: medulloblastoma, atypical teratoid rhabdoid tumors, primitive neuroectodermal tumor, pediatric glioblastoma, and diffuse intrinsic pontine glioma [93]. By targeting the CD47–SIRPα immunological checkpoint, glioblastoma development can be inhibited, and the activity of phagocytic, dendritic, and T-lymphocytes were enhanced, promoting the efficiency of tumor cell clearance [95].

#### 2.3.4. CD44

The major HA receptor CD44 exists as several widely distributed isoforms varying in size from 80–250 kDa. CD44s is the standard form, CD44E is the epithelial form, and CD44v includes variant isoforms. CD44 can be substituted with all major classes of GAGs, while HS can be substituted with a CD44 that has been termed epican [96] and is expressed by keratinocytes [97,98]. Differential splicing of 11 variable exons in CD44 leads to 20 isoforms. CD44–HA interactions promote cell proliferation, migration, and invasion in inflammation and cancer progression [99]. Such interactions are complex and may promote or inhibit these processes. CS- and HS-modified CD44 containing the v3 alternative exon encoding the consensus motif SGXG for GAG addition are found in CD44 isoforms v3-10 and v3, 8-10 and occur in many tumors [100,101,102]. Epican is an HS-modified form of CD44 [97]. Besides binding HA, these GAG-modified forms of CD44 can also bind growth factors impacting cellular proliferation in inflammation and cancer progression.

The extracellular domain of CD44 can also bind collagens [98,103], fibronectin, osteopontin, and MMPs, equipping CD44 with biosensory properties conveying cell transductive cues from the ECM. CD44 has roles in cell–cell interactions, cell adhesion and migration, lymphocyte activation and homing, hematopoiesis, and tumour metastasis [104]. CD44 interactions with integrins, osteopontin, collagens, and MMPs have roles in the stabilisation and remodelling of the CNS/PNS ECM [104,105]. CD44 and integrins attached to neural cells generate focal adhesion complexes that generate traction forces and cell spreading, as well as glioma cell invasion [106,107]. The entrapment of HA by CD44 is important in the hydration and space-filling properties of the CNS/PNS ECM, which maintain ionic gradients and hydration and compartmentalization of neural tissues, preserving neural niches and stem cell viability [108,109,110,111]. MMPs can lead to the shedding of CD44 from the cell surface [112], while soluble CD44 antagonizes CD44-mediated cellular binding to HA and can modulate or inhibit tumor development [113] and tumor cell migration [114]. The transmembrane and cytoplasmic domains of CD44 have interactive properties with the cytoskeleton, as well as roles in signal transduction, which can up-regulate invasive tumor phenotypes and metastasis, leading to matrix degradation and cancer and tumor cell migration [115]. Figure 1 depicts the diverse range of the modular organization of cell-associated HS–PGs.

### 2.4. Functional HS/CS Dual Modification in PGs Controls Cellular Behavior

#### 2.4.1. HS/CS Side Chains and Syndecan Core Proteins Promote Midkine and Pleiotrophin Binding and Tissue Growth

CS chains of SDC-1 and 4 ectodomains comprise nonsulfated, 4-*O*-, 6-*O*-, and 4,6-*O*-disulfated N-acetylgalactosamine disaccharides. SDC-1 and 4 CS chains are significantly different, with a higher degree of sulfation evident in SDC-4 [82]. FGF2 binds to HS chains on SDC-1 and 4, as does midkine (MK) and pleiotrophin (PTN). MK and PTN also bind to CS chains. A stronger binding of MK and PTN has been observed in SDC-4 compared to SDC-1. The removal of CS chains decreases association and dissociation rate constants for MK, PTN, and bFGF in both SDCs, indicating that growth factor binding requires HS and CS chains. The core protein of SDC-1 also contributes to MK and PTN binding to growth factor and the delivery of cell-surface receptors [82]. HS and CS thus display supportive cooperative effects in growth factor binding in the SDCs. This effect is coordinated by interactions with their core proteins.

#### 2.4.2. Co-Ordinated Actions of HS and CS in Neural Cell Regulation

CS has neuronal growth-promoting and inhibitory properties, which depend on CS sulfation, as well as cell and tissue contexts. However, HS universally supports neuronal development [116,117]. Unlike CS, where sulfation is an intracellular process, HS also undergoes extracellular modification by the 6-*O*-sulfatases Sulf1 and Sulf2 [118], which alters interactions with proteins and can promote cell signaling [119]. Two major classes of neuronal receptors bind HSPGs and CSPGs: Type IIa receptor protein tyrosine phosphatases (RPTPσ, RPTPδ and LAR) and the Nogo receptors (NgR1 and NgR3) [120]. HSPG sulfation is crucial to the regulation of synaptic development and neurotransmission. This is achieved by bidirectional control of HS 6-*O*-sulfotransferase (hs6st) and Sulf1 activity [117]. CSPGs generally suppress neurite growth by blocking or interfering with neurite regulatory pathways, integrin, semaphorin, and EGFR pathways using intracellular processes involving calcium, RhoA/ROCK, Akt, PKC, and MAPK signaling to alter microtubule or actin cytoskeleton organization, gene expression, and protein synthesis. RPTPσ and LAR bind to CSPGs with an affinity in the nanomolar range; binding is disrupted by pre-treatment with Chondroitinase ABC, demonstrating that this interaction is due to the CS–GAGs [121,122]. In addition to binding HSPGs, RPTPσ and LAR are functional receptors for CSPGs and can bind neurocan and aggrecan. These interactions are sulfation-dependent; RPTPσ can bind CS-D, CS-E, and DS, but not the more commonly found CS-A or CS-C [117,123]. The CNS/PNS contains a diverse range of CSPGs with instructive roles in the development of embryonic neural axonal networks, and in responses displayed by neural cell populations in mature tissues to traumatic injury [124,125]. Following brain trauma and spinal cord injury, a protective stabilizing CSPG-rich scar tissue is laid down at the defect site. The binding of HS and CS to RPTPσ and LAR mediate opposite effects on neural development and, thus, represent a molecular switch controlling neuronal behavior [126].

#### 2.4.3. HS Acts as a Molecular Switch over Cellular Behaviour in Specific Tissue Contexts

HS and CS–PGs regulate numerous cell surface-signalling events, typically producing opposing effects on cell regulation. CS–PGs inhibit nerve regeneration through RPTPσ decorated with CS. However, HS substitution on RPTPσ and crystallographic analyses of HSPG–CSPG binding sites on RPTPσ reveal these can accommodate diverse GAGs with comparable affinities. HS induces RPTPσ oligomerization in a solution, which is inhibited by CS. RPTPσ HSPGs share a punctate colocalization in sensory neurons in vitro, contrasting with CSPGs, which are distributed throughout the ECM. This has led to a proposal that HS and CSPGs can exert opposing effects on neuronal extension by controlling the oligomerization of RPTPσ. A synthetic HS polymer has subsequently been used to repair the corneal epithelium, and innervation can be regenerated by this polymer, restoring corneal transparency and downregulating myofibroblast activity, as well as scarring after experimental corneal injury [127]. Perlecan can either prevent or promote tumour development. Breast, prostate, lung, and renal cancers all preferentially metastasize to bone in a dense perlecan-rich environment. However, perlecan can also promote the production of MMPs, SULFs, and heparanase in the tumor stroma, modifying the action of perlecan due to the modification of its HS side chains and/or its modular core protein components. These modifications can detrimentally affect cell adhesion, invasion, angiogenesis and tumor development. Thus, perlecan acts as a molecular switch over tumor development [128,129,130].

#### 2.4.4. HS Directs Formation of Neural Networks through Axonal Guidance Proteins Slit and Robo Consistent with Its Roles in Neuritogenesis

Roundabout 1 (Robo 1) is a cell surface-signaling molecule that is important in axonal guidance during neural network formation (Figure 2). The interaction of Robo-1 with HS and members of the Slit protein family confers its axonal guidance activity [131]. O-sulfate groups on HS chains have critical roles in interactions with Slit. Robo 1 interacts with the HS tetrasaccharide IdoA–GlcNS6S–IdoA2S–GlcNS6S–(CH_2_)_5_NH_2_ [131]. Slit–Robo interactions are modulated by HS, while the HS chains of GP1 promote the dimerization of Slit and the formation of a complex with Robo-1, aiding in axonal guidance in neural network formation [55,132,133]. Inhibitory cues from CS also have roles in the regulation of axonal guidance. Robo-1 LRR interactivity aids in its axonal guidance properties. Robo-1 is also a cell adhesion receptor with roles in neuron development and regulates neuronal gene expression [134]. Robo–Slit interactions initiate a cytoplasmic signaling pathway, resulting in the collapse of the local actin and microtubule cytoskeleton and growth cone, resulting in repulsive effects on axonal growth and neural network formation [135]. Once Robo and Slit axon guidance proteins attach to the neuron growth cone cell surface, they respond to their respective ligands, initiating complex cell-signaling events, resulting in the rearrangement of the growth cone, actin, and microtubule cytoskeleton to effect the regulation of axonal growth and axonal guidance [136].

## 3. Basement Membrane HS–PGs

### 3.1. Collagen XVIII

Collagen XVIII is a 300 kDa PG containing three HS chains on a 120 kDa core protein, and it is also a member of the multiplexin collagen family [137]. Collagen XVIII has alternatively spliced long, intermediate, and short isoforms [138,139]. The long isoform contains N-terminal frizzled-like (Fz) and a unique heavily glycosylated DUF959 domain of unknown function. The Fz domain has Wnt-inhibiting activity [140]. Collagen XVIII also contains N-terminal laminin-G like TSP-1 modules. In addition, C-terminal regions also contain a trimerization and a hinge region susceptible to protease cleavage, which releases a terminal 20–28 kDa endostatin anti-angiogenesis peptide module.

### 3.2. Agrin and Perlecan

Agrin, a 400 kDa PG (220 kDa core protein), and perlecan, an 800 kDa (467 kDa core protein) PG, confer functional properties to tissues and instructional control over resident cell populations and roles in the assembly, as well as the stabilization of basement membranes and ECM structures, such as the blood brain barrier (BBB) and neuromuscular junction (NMJ) [141,142,143,144]. Agrin and perlecan bind a range of HS-interactive growth factors, while morphogens regulate cell proliferation, differentiation, and ECM stabilization [14,145,146,147,148]. In cartilaginous tissues [149,150,151,152,153,154], these regulate tissue growth and homeostasis [149,155,156,157], as well as ECM remodelling in development and tissue repair [158,159,160]. Perlecan is a component of progenitor stem cell niches [161,162,163,164,165], where it sequesters FGF-2 [162,163] and promotes stem cell viability, proliferation and differentiation, and the attainment of pluripotency. Agrin has roles in mechanotransductive cross-talk between LRP4–MuSK and integrin-focal adhesion pathway Yes-associated Protein (YAP), which signals through the Hippo pathway [166,167]. Agrin controls the motor neuron stimulation of the LRP4–MuSK receptor in muscles in the NMJ-regulating neuromuscular control. Cardiomyocytes respond to tissue stiffness changes in infarcted cardiac tissue through mechanotransductive effects mediated by Hippo signaling [168]. Agrin YAP promotes the proliferation of epicardial cells [169]. Agrin-associated transcriptional co-activators YAP/TAZ also have roles in the Hippo-mediated homeostatic regulation of alveolar bone mineralization [170] and long-term osteochondral regeneration [158]. Mechanical stress (compression/shear/tension) regulates cartilage development and the maintenance of tissue composition and optimal functional performance [171,172]. The PCM surrounding chondrocytes not only absorbs dynamic and static forces, which are cytoprotective, but it also facilitates mechanotransduction between the PCM and ECM [173], as well as the cyto-protection of chondrocytes and IVD cells [150,174].

### 3.3. Neurexins

The neurexins exist as alternatively spliced isoforms (α, β, γ) and are transmembrane HS–PGs that stabilize the synapse and provide specificity to synaptic interactions. Variation in the HS sequence may also confer specificity to neurexin interactions [175,176]. α-Neurexin core protein contains six laminin G-sex hormone-binding globulin (LNS) domains interspersed with epidermal growth factor like (EGF) domains. β-neurexins contain a single LNS domain and no EGF domains. Presynaptic neurexins and Type IIa protein tyrosine phosphatases (RPTPs) interact with a range of postsynaptic ligands to provide synaptic stabilization [177,178]. Neurexin HS interactions with leucine-rich-repeat transmembrane neuronal proteins (LRRTMs) induce presynaptic differentiation and synaptogenesis [176,179]. Neurexin–neuroligin interactions further promote synaptic development and function. However, incorrect interactions can lead to autism and schizophrenia [180]. Presynaptic neurexin interactions with postsynaptic ligands modulate synaptic input/output signals generated in the synaptic cleft. Neuroligins are major post-synaptic neurexin interactive adhesion molecules [179]. Mutations in genes encoding the neurexins and their ligands have been observed in neuropsychiatric disorders such as schizophrenia, autism, epilepsy, and Tourette syndrome, showing the central roles neurexins play in the control of synaptic plasticity and function [181,182,183,184,185,186,187,188,189,190,191].

### 3.4. Pikachurin

Pikachurin is a 110 KDa retinal basement membrane, HS–PG, that interacts with α-dystroglycan (DG) to stabilize ribbon synapse interactions in photoreceptors with bipolar neurons in retinal neural networks that form part of the phototransductive signals that are transferred in neural transduction during the visual process [119]. The binding of pikachurin to DG requires glycosylation and divalent cations. An incorrect interaction between pikachurin and DG has been noted in muscular dystrophies often associated with eye abnormalities [192,193]. Pikachurin contains multiple Lam G and tType III fibronectin repeats and EGF, which contribute to its protein interactive properties; its LamG domains interact with α-DG [194,195].

### 3.5. Eyes Shut

Eyes shut (Eys) is a modular 250–350 kDa retinal basement membrane HS–PG that is closely related to agrin and perlecan. Eyes shut has roles in the stabilization of the axenome primary cilium that links the upper and lower regions of the photoreceptor rods and cones and ECM remodeling in the developing retinal epithelium [30]. Eys contains multiple EGF and LamG modules and a central Serine–Threonine rich module containing multiple HS chains. Mutations in the *Eys* gene are a common cause of autosomal recessive retinitis pigmentosa (arRP) in Chinese and Japanese sub-populations [196], yet the role of Eys in humans remains to be fully determined [197,198]. Structural depictions illustrating the biodiverse functional forms of extracellular and synaptic HS–PGs are summarized in Figure 3, and their functional properties are listed in Table 1.

## 4. Regulatory Roles for HS–PGs in the Stem Cell Niche Determine Stem Cell Viability and Attainment of Pluripotent Migratory Stem Cell Lineages

Stem cell niches and the differentiation of defined progenitor cell lineages are highly influenced by HS sulphation motifs. Thus, it is unsurprising that the niche environment is also a rich source of glycosyl transferase and sulfotransferase enzymes that modulate the HS expression patterns of progenitor cell PG populations [10,161,253,254,255]. The HS chains on HS–PGs are not synthesised in a template system but are reliant on the localization of specific glycosyl sulfotransferase and extension enzymes to undertake these biosynthetic steps [256]. Thus, the fine structure of HS varies depending on the localization of these enzymes. Perlecan promotes progenitor cell viability in the niche environment, as well as the proliferation, differentiation, and development of cartilaginous tissues [165]. Specific HS glycoforms have critical roles in determining the pluripotency of embryonic stem cells and the development of migratory stem cell lineages that can participate in tissue development.

## 5. HS Interactive Proteins

Several excellent reviews on extracellular and cell surface HS–PGs exist in the literature. The interested reader is referred to these for further information [257,258,259,260,261,262,263,264,265]. In this review, the more important aspects of HS interactive components with representative examples of HS–PGs, such as perlecan and the syndecan, will be covered. HS-substituted proteins (the heparanome) [20,266,267] have cell-directive cell-regulatory properties and ECM-organizational and functional properties. HS–PGs also regulate several essential physiological life processes. Table 2, Table 3 and Table 4 list a selection of these for illustrative purposes. Attempts are being made to develop a broad coverage computational interactome for the entire human proteome using AI methodology [268]. HS proteins are expected to be key components in this database. In a study conducted in 2019, a number of additional HS-binding proteins (HSBPs) were identified in a murine model of the actively inflamed pancreas, increasing the total number of HSBPs to 786 [23,269]. SDC4 has an important role(s) in the insulin secretory response in this model. However, this response does not appear to be mediated purely through SDC GAG chains but is also mediated through core protein interactions [270]. The biodiverse interactive and functional nature of perlecan and the syndecans is clearly evident from the data presented in Table 2 and Table 3.

### 5.1. Perlecan

Perlecan is a modular, multifunctional proteoglycan interactive with a large number of ligands (Table 2). Perlecan has major roles in tissue development, ECM assembly, and stabilization, and it plays roles in specialized structures such as basement membranes, the blood-brain barrier, neuromuscular junction, and blood vessels [250]. Perlecan is a prominent component of foetal rudiment cartilage stem cell niches, and it promotes the differentiation of chondroprogenitor stem cells with roles in diarthrodial joint development [165,271]. Perlecan is an early chondrogenic marker [154,157,272] and promotes the development of rudiment cartilage [273], a transient tissue that is eventually replaced by bone by endochondral ossification as part of the process of skeletogenesis and may have roles in cartilage repair processes in OA [151,159,274,275]. Perlecan sequesters a number of growth factors and has fundamental roles to play in the promotion of the proliferation and differentiation of many cell types to promote tissue development and repair in disease [172]. Perlecan also has mechanosensory properties in weight and tension-bearing connective tissues and acts as a shear flow biosensor in blood and the cannalicular fluid of bone [276], providing instructive cues to endothelial cells and osteocytes, which regulate blood pressure and the metabolism of smooth muscle cells and the laying down of bone as part of the normal extension of the axial and appendicular skeleton [173,174]. Perlecan’s multifunctional properties have led to its proposal as a potential candidate of application in repair biology [160,277], as well as general health and well-being [278]. Perlecan has potential roles in neural stem cell niches [164]; it improves vascular repair in spinal cord injury [279] and neurologic disease [280]; and it has roles in the development of Alzheimers Disease (AD) [281].

**Table 2 ijms-24-14101-t002:** Perlecan-interactive ligands (data compiled from [145,250,282,283,284,285,286,287,288]).

Domain I	Domain II	Domain III	Domain IV	Domain V
Laminin-1	VLDL	FGF-7, 18	Nidogen-1, 2	Nidogen-1
Collagen, IV, V, VI, XI	LDL	FGFBP	Fibronectin	Fibulin-2
Fibronectin	Fibrillin-1	WARP	Collagen IV	β1-integrin
PRELP, WARP	Wnt	Collagen VI	PDGF	α-DG
Fibrillin-1		Tropoelastin	Fibulin-2	FGF-7
Thrombospondin			Collagen VI	Endostatin
FGF1, 2, 7, 9, 10, 18			Tropoelastin	ECM-1
BMP-2, 4			NG2/CSPG4	Collagen VI
PDGF, VEGF, IL2				Progranulin
Hh, Ang-3				Acetyl Ch
Heparanase				α2β1 integrin
Activin A, HistoneH1				Tropoelastin
G6b-B-R				NG2/CSPG4

Abbreviations: CSPG4, melanoma-associated chondroitin sulfate proteoglycan, or neuron-glial antigen 2; PRELP, proline/arginine-rich end leucine-rich repeat protein, Prolargin; WARP, von Willebrand Factor A domain-related protein; FGF, Fibroblast growth factor; BMP, Bone morphogenetic protein; PDGF, Platelet derived growth factor; VEGF, Vascular cell endothelial cell growth factor; α-DG, alpha dystroglycan; IL, Interleukin; Ang-3, Angiopoietin like protein-3; G6b-B-R, Megakaryocyte lineage-specific immunoreceptor tyrosine-based inhibition motif–containing receptor; ECM-1, ECM protein-1; Acetyl Ch, acetyl cholinesterase.

### 5.2. Syndecan

The syndecan family has four members that have diverse interactive properties resembling perlecan in terms of the ECM components it interacts with and the range of receptors, morphogens, growth factors, and cytokines it regulates in a wide range of physiological processes (Table 3).

**Table 3 ijms-24-14101-t003:** Syndecan interactive ligands (compiled from [77,79,133,289,290,291,292,293,294,295,296,297,298,299,300,301,302,303,304,305,306,307,308,309,310,311,312,313,314,315,316,317,318,319,320,321,322,323,324]).

ECM Proteins	Proteases	Integrins Receptors		Morphogens Growth Factors	Cytokines Angiogenic Peptides
Laminins	MMP-2, 7, 9	αVβ3	EphB4	Activin Amphiregulin	GM–CSF
Fibronectin	ADAMTS-4	αVβ5	IGF1R	BMP-2, 4 HB-EGF	IL-2, 3, 4, 7, 12
TSPs	MT1-MMP	α6β4	FGFR	Chordin Neuroregulin	IFN
Collagens	Leucocyte elastase	α2β1	ErbB2	SHH FGF 1–23	TNFα
Fibrin	Cathepsin G	α3β1	CD148	Frizzled proteins PDGF	C–C Chemokine
HB–GAM	Carboxypeptidase	α6β4	L-Selectin	Wnts 1–3 GDNF	CXC
Tenascin	Thrombin	α4β1	P-Selectin	VEGF	Angiostatin
Fibrillin	Plasmin	αMβ2	E-Selectin	HGF	Endostatin
Tropoelastin		NCAM		TGFβ-1, TGFβ-2	Endorepellin
		PECAM			
		EGFR			
		VEGFR2			

Abbreviations: TSPs, Thrombospondins; HB-GAM, Heparin-binding growth-associated molecule; also known as pleiotrophin; MMP, Matrix metalloprotease; ADAMTS-4, A Disintegrin and Metalloproteinase with Thrombospondin motifs; MT1-MMP, Membrane type-1 matrix metalloprotease; EphB4,Ephrin type-B receptor 4; IGF1R, Insulin-like growth factor-1 receptor; FGFR, Fibroblast growth factor receptor; ErbB2, Receptor tyrosine-protein kinase erbB-2, also known as HER-2 (Human epidermal growth factor receptor 2) and CD340; NCAM, Neural cell adhesion molecule; PECAM, Platelet endothelial cell adhesion molecule, also known as cluster of differentiation (CD 31); EGFR, Epidermal growth factor receptor; VEGFR2, Vascular endothelial cell growth factor receptor-2; HB EGF, Heparin-binding EGF-like growth factor; PDGF, Platelet derived growth factor; GDNF, Glial cell line-derived neurotrophic factor; HGF, Hepatocyte growth factor; TGFβ-1, Transforming growth factor-β1; GM-CSF, Granulocyte-macrophage colony-stimulating factor; Il-1, Interleukin-1; IFN, Interferon; TNFα, Tumour necrosis factor-alpha.

The binding thread between perlecan and the syndecan family is the HS side chains, a function-defining component of these PGs. Although each PG also has important core protein modules that provide specific properties, that makes them extremely important bioresponsive effector molecules capable of regulating cellular behavior and physiological processes to maintain optimal tissue properties and tissue homeostasis.

It is beyond the scope of this review to cover the complexities of the fine structure of HS other than to point out the extremely large number of ligands that can interact with HS, which is evident in the entries listed in Table 4. Many excellent reviews have appeared on HS, and the interested reader is directed to these for further information on the diverse roles of HS in various cell and tissue contexts [20,261,325,326,327,328,329,330,331,332,333,334,335,336].

**Table 4 ijms-24-14101-t004:** Examples of the diversity of heparin/HS-interactive proteins.

Molecule	Function, Ligands	Ref.
**Anti-angiogenic agents**		
Angiostatin	38 kDa plasmin fragment derived from plasminogen cleavage by urokinase/tPA, inhibits endothelial cell proliferation, angiogenesis	[337,338,339,340]
Endostatin	20 kDa C-terminal fragment collagen XVIII, anti-angiogenic peptide	[340]
Restin	C-terminal fragment of XV collagen XV, anti-angiogenic peptide	
**Cell adhesion molecules**		
L,E,P-Selectin	Cell adhesion leucocyte homing receptor (CD62) lectin-like sugar binding activity. Expressed by granulocytes, monocytes, lymphocytes, neutrophils	[341,342]
MAC-1	Macrophage-1 antigen complement receptor (CR3) or CD11b	[343]
NCAM	Neurons, glia, skeletal muscle cell adhesion molecule (CD56)	[344,345,346]
PECAM-1	Platelet endothelial cell adhesion molecule (CD31) on platelets, monocytes, neutrophils, T cells, endothelial cells promotes leukocyte transmigration during inflammation, angiogenesis, and integrin activation.	[347,348,349,350]
**Chemokines**		
C-C	Induction of chemotaxis	[351]
CXC	Subfamily of the chemokine superfamily involved in leukocyte trafficking, recruitment, and activation	[352,353,354,355]
RANTES	RANTES is a prototypical T-cell-derived chemokine and potent inflammatory mediator that activates basophils and mast cells and attracts T cells and regulates CD8 T cell responses during chronic viral infection.	[356,357,358]
**Cytokines**		
IL-2, 3, 4, 5, 7, 12	Cytokines associated with innate immunity, trigger inflammation	[359,360,361,362]
GM-CSF	Granulocyte-macrophage colony-stimulating factor, colony-stimulating factor 2 (CSF2), secreted by macrophages, T cells, mast cells, NK, endothelial cells, fibroblasts.	[363,364]
Interferon-γ	Signaling protein released by host cells due to viral infection	[365]
TNF-α	Cell-signaling protein in systemic inflammation and the acute phase response by activated macrophages; regulates immune cells.	[366,367]
G6b-B R	Inhibitory Megakaryocyte-Platelet Receptor G6b-B, regulated by HS, modulates tissue fibrosis modified by platelet function	[287]
PF4	Interacts with HS inhibits AT-dependent thrombin and factor Xa.	[368]
**Redox molecules**		
SOD	Enzyme converting superoxide free radical into O_2_ or H_2_O_2_	[369]
**ECM structural glycoproteins**		
Fibrin	Initial component of clot formation in wound repair	[370]
Fibronectin	High Mw glycoprotein, integrin binding, Cell attachment, ECM organization	[371,372]
Interstitial Collagens	ECM organization/stabilization provided by Type IV collagen stabilizes BM’s, HS binding in Collagen V, XI regulates collagen heterofibril formation, Collagen IV, V, VI, XI interact with perlecan.	[373,374,375,376,377,378]
Laminins	High Mw heterotrimeric ECM and basement membrane component	[379,380,381,382,383,384]
Tenascin	ECM glycoprotein, stabilizes perineuronal net formation in the CNS	[385,386]
TSP-1, 2	Secreted glycoprotein family, anti-angiogenic, matricellular, multifunctional proteins in angiogenesis, apoptosis, TGF-b activation, immune regulation.	[387,388]
Vitronectin	Glycoprotein of hemopexin family found in serum, ECM and bone, binds to αVβ3 integrin to promote cell adhesion and cell spreading.	[389]
collagen V	Regulates collagen heterofibril architecture along with Coll XI	[390,391,392]
collagen XI	Interacts with pericellular perlecan and protects PCM	[376,393]
Histone H1	H1 histones bind dynamically to chromatin in living cells and exchange rapidly between nucleosomes and may regulate transcription.	[394,395]
**Growth factors**		
Amphiregulin	Amphiregullin is an EGF-like ligand that binds to EGFR and is an autocrine growth factor and mitogen for astrocytes, Schwann cells, and fibroblasts.	[396]
Betacellulin	Betacellulin is a member of the EGF growth factor family and an EGFR ligand.	[397,398]
Neuroregulin	EGF family member with diverse roles in neural development, Schwann cell and oligodendrocyte differentiation, binds to, and activates, the ErbB family of RTKs	[399]
Pleiotrophin	18 kDa basic heparin-binding growth factor related to midkine, also known as neurite growth-promoting factor-1, or heparin-binding growth-associated molecule (HB–GAM)	[400,401,402]
Midkine	Heparin-binding growth factor, promotes cell proliferation, migration, angiogenesis, fibrinolysis. It is also known as neurite growth promoting factor-2.	[403,404,405]
FGF Family	22 FGFs, FGF1–10 bind HS-inducing tyrosine kinase cell signaling	[406,407,408]
IGF II	Growth-promoting hormone synthesized in the liver	[409,410,411,412]
PDGF-AA	Potent mitogen for fibroblasts, SMCs, osteoblasts, tenocytes, and glial cells. Stored in the α-platelet granules and by SMCs, activated macrophages, and endothelial cells. Promotes angiogenesis, tissue remodelling, and PI3K-mediated cell differentiation.	[413]
VEGF-165, VEGF-189	Stimulates formation of blood vessels and roles in bone formation, hematopoiesis, wound healing, and tissue development.	[240]
TGF-β1 TGF-β2 TGF-β3	TGF-β is a multifunctional cytokine member of the TGF superfamily occurring as TGF-β1, β2, and β3 isoforms produced by white cell lineages. Key functions include the regulation of inflammatory processes, stem cell differentiation, T-cell regulation, and differentiation. Multifunctional homodimers interactive with Small Latent Complex (LCC), forming a Large Latent ECM Complex with LTBPs; requires proteolytic activation in situ; is anabolic and chondrogenic. HS potentiates TGF-β during wound contraction.	[414,415]
Activins	Interact with HS chains of cell surface and matrix HS–PGs.	[416]
**Growth factor-binding proteins**		
IGFBP-3, 5	Bind IGF-I, II, and cell surface proteins initiating outside-in cell signaling. Overexpressed in pulmonary disease, leading to excessive ECM deposition/fibrosis.	[417]
TGF-β BP	Latent ECM forms of TGF-β laid down in ECM as LTBP 1–4.	[418,419]
Follistatin	Activin-binding glycoprotein and widespread cellular distribution. Regulation/inactivation of TGF-β superfamily members, activin.	[420]
**Proteases/protease Inhibitory proteins**		
AT	Heparin increases the affinity of AT for Factor IIa (Thrombin) and Factor Xa, and significantly increases AT’s inhibitory activity. HS inactivates ATs target enzymes, Thrombin, Factor Xa, and Factor IXa.	[421]
TFPI	TFPI, tissue factor pathway plasma Kunitz serine protease inhibitor anticoagulant, produced by endothelial cells, inhibits Factor VIIa, and Xa prevents tissue factor activation of coagulation cascade.	[422,423]
Factor Xa	Fondaparinux HS pentasaccharide specifically targets Factor Xa in clinical settings such as deep vein thrombosis and cardiac surgery.	[424,425,426,427]
Thrombin	HS anti-coagulant inhibits thrombin activity, preventing clots.	[428,429,430,431]
HNE	HS inhibits elastase activity through electrostatic interaction.	[432]
Cathepsin G	HS inhibits cathepsin-B through electrostatic interaction.	[433]
Chymase	Complexed by HS side chains of granular Mast cell Serglycin.	[67,434]
TIMP-3	A and B β-strand N-terminal domain TIMP-3 hep-binding	[435,436,437]

Abbreviations: tPA, tissue plasminogen activator; CCR, β-chemokine receptor; CCL, β-chemokine ligand; IL, interleukin; TNF, tumour necrosis factor; AT, antithrombin; ECM, extracellular matrix; PCM, pericellular matrix; LTBP, latent transforming growth factor-β-binding protein; TRK, tyrosine receptor kinase, LLC, large latent complex; LAP, latency-associated peptide; HBGAM, heparin-binding growth-associated molecule; TGF-β BP, transforming growth factor-β binding protein; TSP, thrombospondin.

## 6. The Tissue-Protective Properties of HS–PGs

Much has been published on the roles of HS–PGs in tissue development and ECM remodelling in wound repair. However, it has not been as readily acknowledged that the HS component of HS–PGs can also have important tissue protective properties by improving the inhibitory efficiency of a number of protease inhibitory proteins (Table 5).

### 6.1. HS Interactions with Serpins Improves Their Inhibitory Efficiency and Tissue-Protective Properties

A number of GAGs, including heparin and HS, are interactive with serpin protease inhibitory proteins. This can improve their tissue-protective properties (Table 5). Several serpins have important roles in hemostasis. For example, AT inhibits many coagulation proteases, including Xa and thrombin, heparin co-factor II (HCII) inhibits thrombin, Protein C inhibitor (PCI) inhibits activated Protein C (APC), and thrombin is bound to Thrombomodulin, while PAI-1 inhibits tPA. In addition, α2-antiplasmin inhibits plasmin; these all have roles in inflammation. These also regulate the vasculature to tumors required for tumor development. Heparin and HS induce conformational changes in Serpins that improve their interactive properties with proteases and has been finely tuned through evolutionary selection pressure to provide high levels of regulatory control [438]. HCII has a unique role in vascular homeostasis, interacting with endothelial DS in the injured arterial wall and providing an antithrombotic effect [439], and it inhibits thrombin rapidly in the presence of DS or heparin. This increases the rate of the inhibition of thrombin over 1000-fold [439]. A computational review of a 46,656-HS hexasaccharide library has identified a rare sequence of consecutive 2-*O*-sulphated GlcA residues that target HCII with minimal activity on the related Serpin AT. This improves HCII activity at least 250-fold [439]. α1-PI and ITI interact with another GAG, HA, and protect it from free radical depolymerisation [440]. This is important since high molecular weight HA has anti-inflammatory protective properties in tissues [441,442] and has found application in tissue engineering [443,444,445]. High molecular weight HA also regulates cell migration and promotes tissue-repair processes [443,445]. In contrast, depolymerized HA is pro-inflammatory, and it induces MMP production and activation in many cell types, resulting in ECM degradation. The ITI inhibitor family has also been shown to transfer and cross-link its heavy chains to HA in a trans-esterification reaction catalyzed by TSG-6, which stabilizes HA in tissues and has specific roles in reproduction [446]. Secretory leucocyte protease inhibitor (SLPI) interactions with heparin improves its inhibitory activity against proteases active in inflammatory conditions in asthma [447]. The inhibition of human mast cell chymase by SLPI is enhanced through its interaction with heparin [448]. Thrombomodulin, a CS–PG, accelerates the inhibition of thrombin and activated Protein C (APC) by a Protein C Inhibitor (PCI) [449]. HS and heparin improve the inhibitory activity of PCI in its interactions with thrombin, APC, factor Xa, urokinase, and chymotrypsin [450]. The inhibitory properties of kallistatin, a pleiotropic serpin with vasodilatory, anti-angiogenic, anti-inflammatory, antioxidant, anti-apoptotic, anti-fibrotic, and anti-tumor activities, are also improved by the interaction with HA, improving its tissue-protective properties. This is achieved by the inhibition of kallikrein activity and by blocking VEGF, TNFα, HMBG1, Wnt, TGF-β, and EGF cell-signaling pathways [451]. This inhibits the development of hypertension, heart and kidney disease, arthritis, sepsis, influenza virus infection, tumor growth, and metastasis [451]. Plasminogen activator inhibitor-1 (PAI-1) has roles in the regulation of angiogenesis and coagulation; its activity is modulated by cofactors such as heparin and HS. Many HS-binding serpins regulate coagulation cascades and are potent anti-angiogenic agents [452]. Protease nexin-1 (PN-1) is an anti-coagulant [453] and has anti-angiogenic properties [454]. Heparin and HS bind AT, HCII, PCI, PAI-1, kallistatin, and α1PI while HCII utilizes DS as a cofactor [455]. PAI-1 inhibits tPA activity and interactions with heparin or HS, which improves this activity [438]. PN-1 is synthesized by several cell types, including astrocytes, smooth muscle cells, and fibroblasts, and it is deposited in the ECM [456], binding to ECM. HS–PGs accelerate PN-1′s inhibition of thrombin. PN-1 also binds to Type IV collagen with no improvement in its anti-thrombin inhibitory properties [457]. However, PN-1 can rapidly inhibit thrombin and can also inhibit urokinase and plasmin.

### 6.2. TIMP-3 GAG Interactions Are Tissue Protective

Heparin regulates MMP2 and TIMP3 protein levels in tissues and MMP2 activity through interactions between the hemopexin domain of MMP2 and the TIMP3-C-terminal region, which results in inhibition of MMP2 and regulates ECM remodeling [458]. Plasmon resonance spectroscopy shows TIMP-3 is a heparin-binding protein. This interaction is chain-length dependent and involves *N*-sulfo and 6-*O*-sulfo groups. Chondroitin sulfate-B and E also exhibit strong binding to TIMP-3 [459]. Such GAG–TIMP3 interactions have important roles in the maintenance of ECM homeostasis [435] and have also been examined to promote tissue repair [460]. TIMP3 inhibits ADAMTS-4. This interaction is promoted by the Chondroitin-6-Sulphate side chains of aggrecan in cartilaginous tissues and the TSP-1 domains of ADAMTS-4 [461]. TIMP-3 also regulates MMP activity, while the processing of cell-surface receptors, chemokines, and cytokines are affected by MMPs. TIMP-3 is a heparin-binding protein with an affinity of ~59 nM. TIMP-3 also displays strong binding with CS–E and CS B but displays weak binding with HS and CS–A, which are often implicated in tissue destruction and disease processes. Therefore, it is important to have components capable of regulating these enzymes effectively as components in the ECM or resident on the surface of cells, which may become damaged.

**Table 5 ijms-24-14101-t005:** GAG–Protease inhibitor interactions improve tissue protective functions.

Inhibitor	GAG	Functional Properties of HS-PG GAG Interactions	Ref.
α1-PI	HA	Protects HA from depolymerisation by ROS during inflammatory conditions/wound repair.	[440]
Bikunin ITI-L chain	HA	Transfer and covalent attachment of ITI H chains by a trans-esterification process catalyzed by TSG-6 and cross-links; stabilizes high Mw HA.	[446]
AT	HS (saccharide)_5_ GlcNAc 3-*O*-SO_4_	Anti-coagulant inhibits thrombin, Factor Xa; IXa inhibits inflammation; angiogenesis aids in TBI and BBB repair, as well as neurocognitive recovery from TBI.	[462]
HCII	HS, DS 2-*O*-SO_4_, GlcA	Contributes to AT activity, inhibits Factor Xa, VIIIa, thrombin.	[455,463,464]
TFPI	HS, DS	Mediates TFPI-2/MSPI binding to Hep/DS to inhibit plasmin, trypsin, chymotrypsin, plasma kallikrein, Cathepsin G, and F VIIa. TFPI binds to cell surface HS–PGs.	[227,423,465,466,467,468]
PCI	HS/Hep	HS/Hep enhance the activity of PCI, accelerating the inhibition of α, γ-thrombin, APC, factor Xa, urokinase, and chymotrypsin.	[450,469]
Kallistatin	HS	Blocks VEGF, TNFα, HMBG1, Wnt, TGF-β, and EGF cell signaling. Inhibits OA, hypertension, heart and kidney disease, sepsis, influenza virus infection, tumor growth, and metastasis.	[451]
PAI-1	HS/Hep	Hep/HS improves PAI-1 activity. Hep inhibits tPA synthesis matrix deposition of PAI-1 by human mesangial cells. Hep/HS enhances synthesis of two chain urokinase inhibitor PAI-1 form.	[77,470,471]
PN-1	HS	Rapidly inhibits thrombin. Its physiological substrate also inhibits urokinase and plasmin, binding to ECM. HS–PGs accelerates inhibition of thrombin. It is anti-coagulant and anti-angiogenic.	[453,454,457,465,472]
SLPI	12–14 unit Heparin oligos	Stoichiometric 1:1 binding of 12–14 Heparin oligosaccharide to SLPI accelerates SLPI inhibitory activity against proteinase-3, neutrophil elastase, cathepsin G, and mast cell tryptase/chymase.	[447]
TIMP-3	Heparin, HS	TIMP-3 regulates the activity of MMPs, ADAMTS4, and ADAMTS5.	[459,473]

Abbreviations: ROS, reactive oxygen species; ITI, inter-alpha trypsin inhibitor; TSG-6, Tumor necrosis factor-(TNF) stimulated gene-6; AT, antithrombin; TBI, traumatic brain injury; BBB, blood brain barrier; TFPI/MSPI, tissue factor pathway inhibitor/matrix-associated serine protease inhibitor, a Kunitz-type serine protease inhibitor; APC, activated protein C; PCI, Protein C inhibitor; VEGF, vascular endothelial cell growth factor; TNFα, tumour necrosis factor; HMBG1, high mobility group box 1; Wnt, an acronym for Wingless and Integrated; TGF-β, transforming growth factor; EGF, epidermal growth factor; PAI-1,plasminogen activator inhibitor; tPA, tissue plasminogen activator; PN-1, protease nexin; SLPI, secretory leucocyte protease inhibitor; TIMP-3, tissue inhibitor of matrix metalloprotease; MMPs, matrix metalloprotease; ADAMTS, a disintegrin and metalloprotease with thrombospondin motifs.

## 7. Conclusions

HS–PGs continue to be identified, and their sophisticated properties in the regulation of cellular behavior and essential physiological processes continue to be uncovered. HS–PGs represent an extremely interesting group of functional proteins and advance the understanding of how these proteins act as functional cell-instructive effector molecules, which is expected to continue to evolve with the identification of new members. This review has outlined examples of the biodiversity and functionality of HS–PGs and the regulatory processes they control. HS is a function defining feature of HS–PGs but does not act in isolation. The molecular switch provided by dually modified cell-surface HS–PGs such as betaglycan and CD47, which are good examples of the activity of HS, which is regulated by CS. Interactive core protein modules in HS–PGs also have biodiverse functional properties. A greater understanding of these processes outlines the potential of development of HS–PGs as therapeutic agents that could potentially be applied to control cellular behavior to maintain tissue homeostasis or to promote wound healing. Proteoglycan biomimetics represent an extremely powerful therapeutic capable of modifying disease processes and has significant potential in repair biology. This will only become possible with a greater understanding of the biology of HS–PGs. Advances in analytical techniques are facilitating a greater understanding of the structure function relationship between the HS side chains of HSPGs. With the continued improvements in these techniques, predictive capabilities are now becoming available to better understand the atomistic contributions of the conformational environment of HS chains and how this effects cell-instructive capability [269]. This is an exciting area in therapeutic development in the search for novel agents with improved capability in repair biology [125,164,474,475,476,477,478]. A greater understanding of how HS–PGs function in their particular tissue niches and cellular contexts holds considerable potential in the development of novel applications in repair biology.

## Figures and Tables

**Figure 1 ijms-24-14101-f001:**
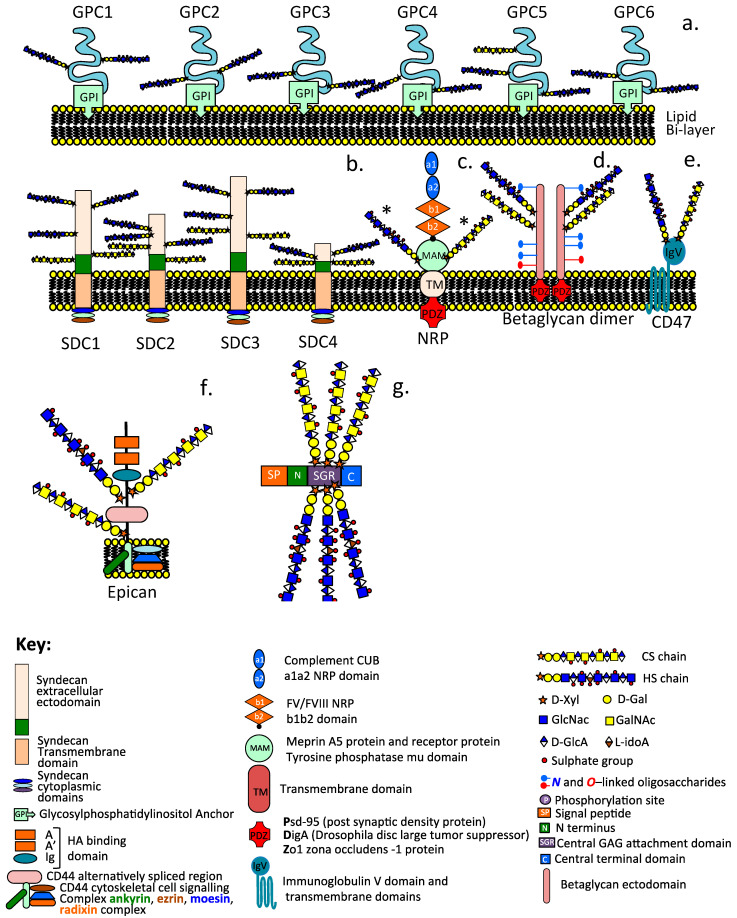
Schematic depictions of the diverse modular structures of intracellular and cell associated HS–PGs including the glypican family (GP1-GP6) (**a**), syndecan family (SDC1-SDC4) (**b**), neuropilin (**c**), betaglycan dimer (**d**), CD47 (**e**), epican, HS substituted form of CD44 (**f**), and the intracellular granular HS-PG serglycin (**g**). The asterisk label on NRP (**c**) indicate that this can contain HS or CS substitution but not both on the same NRP molecule.

**Figure 2 ijms-24-14101-f002:**
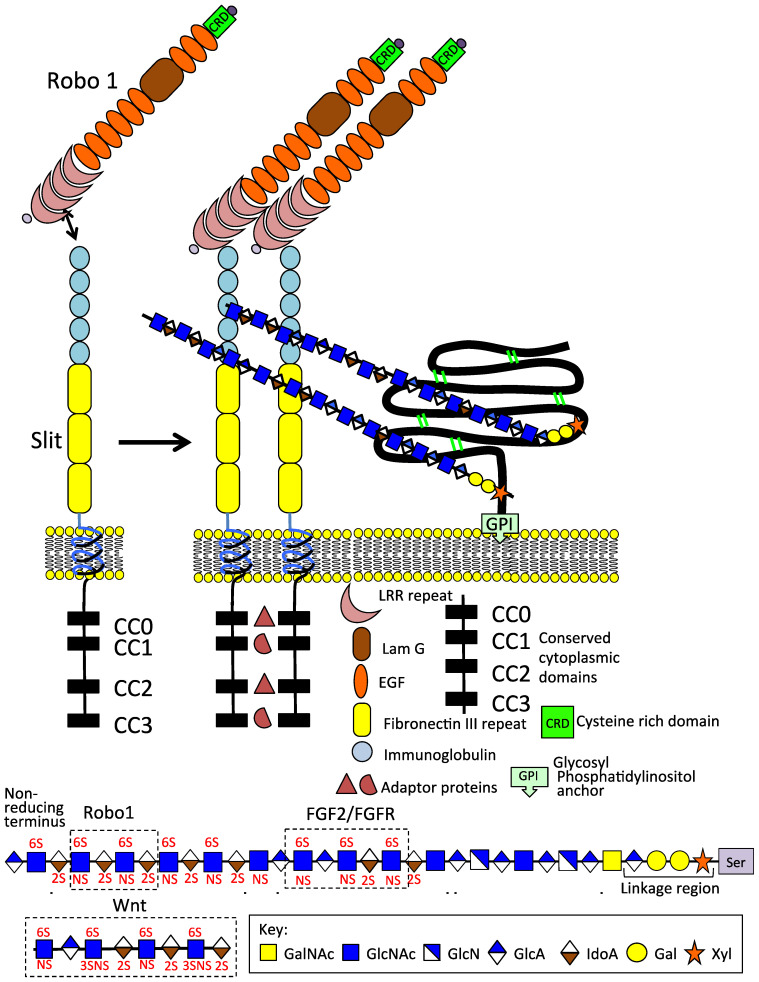
Schematic depiction of the Slit 1 and Robo 1 axonal guidance proteins on the neuron cell surface; their modular structures and GPI anchored glypican1 which promotes Slit 1 dimerisation. The HS chains of GP1 interact with the IgG1 and 2 domains of Slit, 6-*O* HS sulphation is an important interactive determinant. An HS tetrasaccharide has also been identified that interacts with Robo 1. Thus, HS modulates Robo 1 and Slit interactions consistent with its roles in the promotion of neuritogenesis, axonal guidance, and neural network formation. FGF2/FGFR and Wnt HS interactive sequences are also shown.

**Figure 3 ijms-24-14101-f003:**
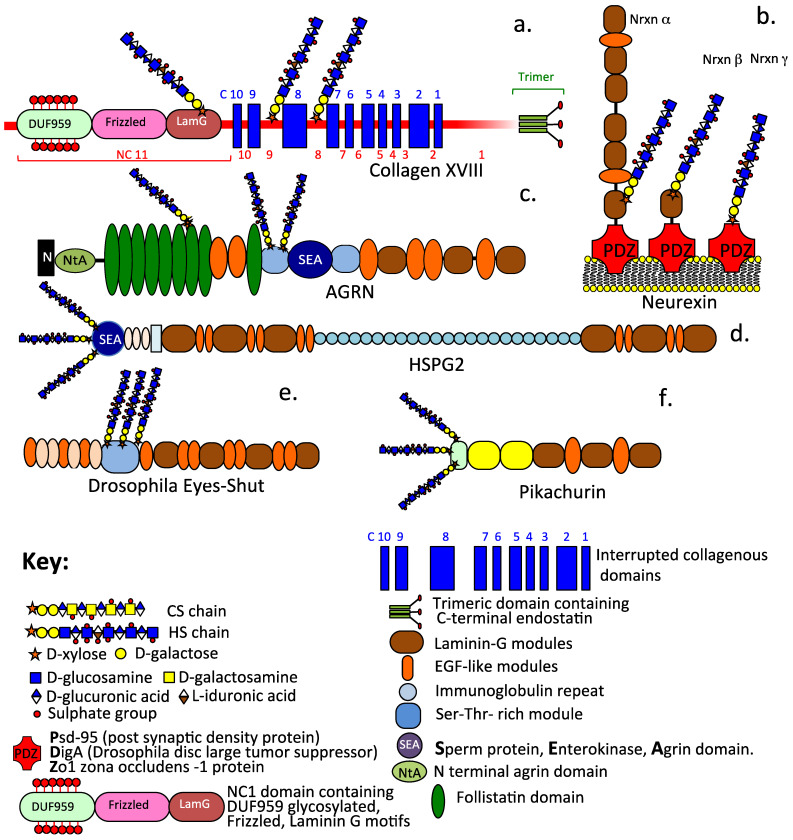
Schematic depictions of the diverse modular structural organization of extracellular and synaptic HS–PGs including collagen XVIII (**a**), the synaptic stabilizing neurexin family of α, β, γ alternatively spliced isoforms (**b**), basement membrane agrin (**c**) and perlecan (**d**), plus the α-dystroglycan interactive proteoglycans Eyes-shut (**e**) and Pikachurin (**f**) which stabilize the axoneme primary cilium and the ribbon synapse of photoreceptors.

**Table 1 ijms-24-14101-t001:** The Biodiversity of HS–Proteoglycans.

Member	Characteristics	Distribution/Function	Ref.
**Intracellular**			
Serglycin	Intracellular granule Heparin, HS, and CS–PG, 17.6 kDa core protein, GAGs are attached to central 16 amino acid SGR domain in SRGN	Expressed by haematopoietic cells, neutrophils, monocytes, lymphocytes, platelets, macrophages, mast cells, megakaryocytes, chondrocytes, endothelial, and embryonic stem cells. Forms inactive protease complexes in intracellular granules	[62,63,64,65,199]
**Cell-associated**			
The Glypicans	GPI anchored HS–PG endocytic co-receptors	Cell surface co-receptors for GFs, ECM proteins, proteases, protease inhibitors, Wnt signaling.	[4,6,69,70,200]
Glypican-1	64 kDa core protein	GPC-1 is expressed in embryonic CNS and skeletal systems and in adult kidney and skeletal muscle. HS side chains facilitate interactions with Wnt/β-catenin, Hh, FGFs, IGFs, VEGF, TGF-β.	[69,201,202,203,204,205,206]
Glypican-2 cerebroglycan	57 kDa core protein	Preferentially expressed in neural tissues, binds Midkine and regulates neural network development.	[207,208,209]
Glypican-3	69 kDa core protein	Roles in mesoderm and intestine, binds FGF2, regulates tissue development, neural network assembly, modulates BMP/FGF mediated renal branching, tissue morphogenesis, Hh signaling.	[4,69,70,116,210,211,212,213,214]
Glypican-4 K-glypican	57 kDa core protein	Found in blood vessels, kidney, brain, promotes CNS development, expressed in the ventricular neuroprogenitor stem cells, committed neurons do not express GPC4, expressed by cortical, hippocampal, cerebellar astrocytes, roles in synapse formation/fore-brain development.	[215]
Glypican-5	84 kDa core protein, a CS chain may modulate HS-GF GPC-5 interactions	Developmentally regulated in kidney, limbs, and brain. Controls cell growth and differentiation.	[216]
Glypican-6	65 kDa core protein	Regulates ERK1/2 signaling neurotransmitter receptor localization in postsynaptic membrane, roles in presynapse assembly and action, broad distribution in fetal kidney, adult ovary, liver, gall bladder.	[217]
Syndecans	Single pass TM PGs containing 3–5 CS/HS chains	Co-receptors of FGFs, VEGFs, TGFβ, AT, Fibronectin	[77,80,218]
Syndecan-1	33 kDa core protein	Found in skin, liver, kidney, lung, and small intestine.	[219,220,221,222]
Syndecan-2 fibroglycan	23 kDa core protein	Found in brain, heart, skeletal muscle, liver, kidney, and lung; regulates TGFβ signaling, angiogenic sprouting.	[223,224]
Syndecan-3 N-syndecan	43 kDa core protein	Found in brain, heart, skin, and skeletal muscle. Colocalizes with TFPI on surface of endothelial and SMCs.	[225,226,227]
Syndecan-4 Ryudocan Amphiglycan	22 kDa core protein	Found in brain, liver, kidney, lung, heart, skeletal muscle, skin, and pancreas. Regulates FGF-2, FAK, ADAMTS-4 activation, and insulin secretion in pancreas.	[225,228,229,230,231,232]
Betaglycan TGF-β-R III	300 kDa Cell-surface CS/HS co-receptor, binds TGF-β via core protein, and FGF-2 via HS.	Multifunctional co-receptor binds TGF-β2, InhA, BMP-9, BMP-10, and FGF-2. MMPs and plasmin shedses release a soluble ecto-domain that antagonizes cell-bound betaglycan activity.	[39,43,44,84,233]
CD44 Epican	Epican is a HS isoform. 80–250 kDa TM HS substituted CD44 HA receptor.	Interacts with osteopontin, collagens, and MMPs; stabilizes/remodebrain ECM and generates focal Interacts with osteopontin, collagens, MMPs, stabilises/remodels ECM. Attaches neural cells to brain ECM, generates focal adhesion complexes/traction forces during cell migration and spreading. Entraps HA and water in brain, preserving ionic gradients and neural cell niches.	[96,97,98,104,105,234,235,236]
CD47	Haematopoietic, 50 kDa HS/CS TM high affinity TSP-1 receptor of CD47-SIRPα system. Protects cells from degradation by macrophages.	HS on CD47 inhibits T cell receptor signaling with TSP-1. Regulates cell migration, proliferation, and vascular cell survival in innate/adaptive immune regulation. TSP-1 inhibits eNOS activation/NO production.	[87,89,90,93,94]
NRP-1	120 kDa single span TM co-receptor, has roles in tumor vascularization in brain, prostate, breast, colon, and lung cancer. Is a SARS-CoV-2 receptor in COVID-19 disease.	Soluble NRP-1 antagonizes cellular NRP-1 activity in axonal guidance, angiogenesis, cell survival, migration, and tumor invasion. Binds HS-binding GFs, VEGF 165, and placenta growth factor; modulates VEGF/VEGFR2 signaling.	[35,38,237,238,239,240,241]
Neurexins α, β, γ	α, β, γ isoforms interact with neuroligin and LRRTM-2, 4 to stabilize synapse and neural-signal transduction.	Provides synaptic stabilization and neuronal plasticity in neural networks. Nrxn dysregulation is implicated in neuro-degenerative cognitive disorders.	[29,181,183]
**Extracellular matrix**			
Collagen XVIII	Homotrimeric multiplexin (300 kDa) with 3 HS chains, in long, medium, and short isoforms; multiple collagenous and NC-domains, 20 kDa C-terminal anti-angiogenic endostatin peptide.	Interacts with laminin, perlecan, fibulin, and nidogen, forming network formations that stabilize basement membranes. LamG and TSP-1 protein interactive domains, and a homologous Frizzled receptor cysteine rich domain, have Wnt inhibitory activity.	[9,139,140,242,243,244,245,246]
Agrin	~220 kDa core protein; HS/CS increases size to 400 kDa. Multiple Lam G and EGF repeats, central Ser–Thr-rich domain contains HS.	Component of cartilage, roles in synaptic organization, and NMJ assembly facilitating regulation of neuro-muscular control and mechanotransduction.	[9,149,155,166,247]
Perlecan	467 kDa core protein with five functional domains. Domain I has 3 HS/CS/KS chains, domain V also has GAG. Domain IV has 23 Ig repeats, Domain V has multiple LamG/EGF repeats and antiangiogenic endorepellin peptide, domain V promotes BBB and angiogenic repair	Ubiquitous HS, or HS/CS PG, sequesters acetylcholinesterase in NMJ. Protects chondrocyte viability. Promotes vascularization, 20 kDa domain V fragment inhibits angiogenesis, and stabilises BM, BBB, BVs, and stem cell niche. HS chains sequester GFs to promote cell proliferation/differentiation, tissue repair, and Domain II LDLR transports Wnt and ShH in tissue development. Reduces SMC proliferation and atherosclerotic plaque deposition.	[112,144,150,151,154,160,163,248,249,250,251]
Eyes-shut	Modular 250–350 kDa retinal PG related to agrin/perlecan; central Ser–Thr module binds HS chains; has multiple EGF and LamG modules.	Organizational roles in the assembly and function of interphotoreceptor ECM in the retinal epithelium, as well as stabilization of ciliary axoneme and ribbon synapse in rods and cones. Vital roles in the photoreceptor ECM stabilization and visual acuity.	[30,252]
Pikachurin	Modular 110 kDa photoreceptor HSPG	Interacts with αDG, stabilises photoreceptor	[31,32,119]

Abbreviations: TGF-β, transforming growth factor beta; InhibA, inhibin A; VEGF, vascular endothelial cell growth factor; BMP, bone matrix protein; GF, growth factor; MT, MMP, membrane type matrix metalloprotease; LamG, LG, laminin G domain; Fibr, fibronectin; NMJ, neuromuscular junction; ECM, extracellular matrix; PG, proteoglycan; LDLR, low density lipoprotein receptor; BBB, blood brain barrier; HS, heparan sulphate; Hep, heparin; CS, chondroitin sulphate.; TSP, thrombospondin; SIRPα, Signal regulatory protein alpha; NO, nitric oxide; eNOS, endothelial cell type nitric oxide synthase; EGF, epidermal growth factor; GAG, glycosaminoglycan; CNS/PNS, central and peripheral nervous systems; GPC, glypican; SDC, syndecan.

## Data Availability

All data are available in individual cited studies in this review.

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
