# Peer review of "The Glycosaminoglycan Side Chains and Modular Core Proteins of Heparan Sulphate Proteoglycans and the Varied Ways They Provide Tissue Protection by Regulating Physiological Processes and Cellular Behaviour"

_ijms, 2023, doi:10.3390/ijms241814101_

Round 1

Reviewer 1 Report

Specific Comments:

1. Section 2.3.3. CD47

Comment: It would be beneficial to a wider audience if the authors can include a brief discussion on the role of CD47 in cancer immune evasion to highlight the importance of HS-PGs in tumor-immune interactions.

2. Section 2.3.4 CD44

Comment: It would be beneficial to a wider audience if the authors can include a brief discussion on the role of CD44 in cancer progression, specifically, through ECM signaling mediated via hyaluronic acid and ECM stiffness to highlight the importance of HS-PGs in tumor-ECM interactions.

3. Section 3.5. Eyes shut; Lines 224 - 226; ‘………..functional forms of HS-PGs are summarised in Figure 1 and the properties of HSPGs listed in Table 1.’

Comment: It will be beneficial to the reader if authors cite Figure 1 and Table 1 in the text wherever relevant HS-PGs (‘Members’) are discussed.

Minor comments:

1. Section 1. Introduction; Lines 55-63; ‘In a comprehensive review of proteoglycan nomenclature conducted in…………… retinal neural networks essential for high quality ocular vision.’

Comment: It would be beneficial if the authors can break the sentence since it is too long and doing so will improve the readability.

2. Comment: At multiple places in the manuscript, spiral-shaped special characters seem to appear in an erroneous manner (For example, Lines 82, Table 1. Betaglycan and Neurexins sections). It would be beneficial if authors revise the manuscript to remove those special characters. 

3. Section. 2.1. cytoplasmic HS-PGs

Comment: ‘c’ in cytoplasmic should be capital.

4. Section 2.3.3 CD47. AND Section 3.2. Agrin and Perlecan. AND Section 5. HS Interactive proteins.

Comment: Period should be removed after the section title.

5. Section 3.3. Neurexins, Lines 194 - 196, ‘Presynaptic neurexins and type IIa protein tyrosine 194 phosphatases (RPTPs) interact with a range……………’

Comment: Reference is missing.

6. Figure 1. Schematic depictions of the diverse modular structural organization of HSPGs

Comment: The resolution of Figure 1 needs to be improved.

7. Table 1. The Biodiversity of HS-Proteoglycans.

Comment: Table 1 needs to be reformatted to improve the demarcation between the ‘Characteristics’ and ‘Distribution/Function’ columns.

8. Comment: The manuscript should be revised to fix the uneven spacing between sentences throughout the manuscript.

9. Comment: Commas are missing at multiple places and the manuscript should be revised to resolve such minor grammatical errors.  

The article is generally well-written and well-articulated. However, the article should be revised to address minor grammatical errors to improve its readability as indicated in the minor comments. 

Author Response

Point by Point Responses to reviewers.

Reviewer 1 comments

Major segments in the revised MS are highlighted,a marked up copy of the revision is attached

Top of Form

  1. Section 2.3.3. CD47

Comment: It would be beneficial to a wider audience if the authors can include a brief discussion on the role of CD47 in cancer immune evasion to highlight the importance of HS-PGs in tumor-immune interactions.

Author response

The following segment has been added to the CD47 section in the revised manuscript.

CD47 is a widely expressed integrin associated transmembrane protein encoded by the CD47 gene and is a member of the immunoglobulin superfamily that binds to integrins, thrombospondin-1 and signal-regulatory protein alpha (SIRP), a protein expressed by macrophages and dendritic cells [104]. Upon binding to CD47, SIRPα initiates a signaling cascade that results in the inhibition of phagocytosis [103]. Phosphorylation of the immunoreceptor tyrosine-based inhibition motifs present on the cytoplasmic tail of SIRPα initiates this response [105]. Subsequent binding and activation of SHP-1 and SHP-2 [src homology-2 (SH2)-domain containing protein tyrosine phosphatases] blocks phagocytosis, by preventing accumulation of myosin-IIA at the phagocytic synapse [106-109]. CD47 is a critical regulator of innate immune surveillance, the HS chain of CD47 is necessary for inhibition of T cell receptor signaling by thrombospondin-1 [110]. CD47 has important roles in the regulation of cellular migration, proliferation, and survival of vascular cells in innate and adaptive immune regulation [98, 105]. TSP-1 acts via CD47 to inhibit NO signaling in the vascular system and supports blood pressure regulation by limiting eNOS activation and endothelial-dependent vasorelaxation [106]. CD47 -SIRPα interactions represent a cell-cell communication system [102, 103, 111] termed an innate immune checkpoint sending a "don't eat me" signal to macrophages [108]. This allows tumor cells to avoid detection and be cleared by the immune system. Blockade of the CD47-SIRPα interaction using humanised antibodies to CD47 (Hu5F9-G4) has yielded promising results in a number of human malignancies including pediatric brain tumors: medulloblastoma, atypical teratoid rhabdoid tumors, primitive neuroectodermal tumor, pediatric glioblastoma, and diffuse intrinsic pontine glioma [107]. By targeting the CD47-SIRPα immunological checkpoint glioblastoma development can be inhibited and the activity of phagocytic, dendritic and T-lymphocytes enhanced promoting the efficiency of tumor cell clearance [109, 112-114]. 

  1. Section 2.3.4 CD44

Comment: It would be beneficial to a wider audience if the authors can include a brief discussion on the role of CD44 in cancer progression, specifically, through ECM signaling mediated via hyaluronic acid and ECM stiffness to highlight the importance of HS-PGs in tumor-ECM interactions.

Author response

The following segment has been added to the CD47 section in the revised manuscript.

The major HA receptor CD44 exists as several widely distributed isoforms varying in size from 80-250 kDa. CD44s is the standard form, CD44E the epithelial form and CD44v variant isoforms. CD44 can be substituted with all major classes of GAGs, HS substituted CD44 has been termed epican [115] and is expressed by keratinocytes [116, 117]. Differential splicing of 11 variable exons in CD44 leads to 20 isoforms. CD44-HA interactions promote cell proliferation, migration, and invasion in inflammation and cancer progression [118]. Such interactions are complex and may promote or inhibit these processes. CS and HS modified CD44 containing the v3 alternative exon encoding the consensus motif SGXG for GAG addition are found in CD44 isoforms v3-10 and v3,8-10 and occur in many tumors [119-121]. Epican is an HS modified form of CD44 [116]. Besides binding HA these GAG modified forms of CD44 can also bind growth factors impacting on cellular proliferation in inflammation and cancer progression.

The extracellular domain of CD44 can also bind collagens [117, 122] fibronectin, osteopontin and MMPs equipping CD44 with biosensory properties conveying cell transductive cues from the ECM. CD44 has roles in cell-cell interactions, cell adhesion and migration, lymphocyte activation and homing, hematopoiesis, and tumour metastasis [123]. CD44 interactions with integrins, osteopontin, collagens, and MMPs have roles in the stabilisation and remodelling the CNS/PNS ECM [123, 124]. CD44 and integrins attached to neural cells generate focal adhesion complexes that generate traction forces and cell spreading and glioma cell invasion [125, 126]. Entrapment of HA by CD44 is important in the hydration and space-filling properties of the CNS/PNS ECM maintaining ionic gradients and hydration and compartmentalisation of neural tissues preserving neural niches and stem cell viability [127-130]. MMPs can lead to shedding of CD44 from the cell surface [131], soluble CD44 antagonises CD44 mediated cellular binding to HA and can modulate or inhibit tumour development [132] and tumor cell migration [133]. The transmembrane and cytoplasmic domains of CD44 have interactive properties with the cytoskeleton and roles in signal transduction which can up-regulate invasive tumor phenotypes and metastasis leading to matrix degradation, cancer and tumor cell migration [134]. Figure 1 depicts the diverse range of the modular organisation of cell associated HSPGs.

  1. Section 3.5. Eyes shut; Lines 224 - 226; ‘………..functional forms of HS-PGs are summarised in Figure 1 and the properties of HSPGs listed in Table 1.’

Comment: It will be beneficial to the reader if authors cite Figure 1 and Table 1 in the text wherever relevant HS-PGs (‘Members’) are discussed.

Minor comments:

  1. Section 1. Introduction; Lines 55-63; ‘In a comprehensive review of proteoglycan nomenclature conducted in…………… retinal neural networks essential for high quality ocular vision.’

Comment: It would be beneficial if the authors can break the sentence since it is too long and doing so will improve the readability.

Authors response: section has been separated into two sentences to improve readability. Section now reads:

In a comprehensive review of proteoglycan nomenclature conducted in 2015 23 a total of 22 HS-PGs were identified, additional HS-PGs have since been identified including the neurexin α, β, γ family with roles in synapse stabilization, synaptogenesis, synaptic plasticity and neural function . Further, two retinal basement membrane HS-PGs, Eyes-shut 28 and Pikachurin 29,30 have also been identified with roles in the stabilization of the rod and cone photoreceptor axenome primary cilium, inter photoreceptor ECM, and the photoreceptor ribbon synapses which interact with bipolar neurons facilitating photo-transduction and neurotransmission in retinal neural networks essential for high quality ocular vision.

  1. Comment: At multiple places in the manuscript, spiral-shaped special characters seem to appear in an erroneous manner (For example, Lines 82, Table 1. Betaglycan and Neurexins sections). It would be beneficial if authors revise the manuscript to remove those special characters. 

Authors response: Spiral symbols have been removed from the following lines:

  • Line 84
  • Table 1 with reference to TGF-β
  • Table 1 with reference to Neurexins α, β, γ

  1. Section. 2.1. cytoplasmic HS-PGs

Comment: ‘c’ in cytoplasmic should be capital. Done

Authors response: Text has been amended to 2.1 Cytoplasmic HS-PGs

  1. Section 2.3.3 CD47. AND Section 3.2. Agrin and Perlecan. AND Section 5. HS Interactive proteins.

Comment: Period should be removed after the section title.

Authors response: Text has been amended.

  1. Section 3.3. Neurexins, Lines 194 - 196, ‘Presynaptic neurexins and type IIa protein tyrosine 194 phosphatases (RPTPs) interact with a range……………’

Comment: Reference is missing.

Author response

The following references have been cited

Uchigashima M, Hayashi Y, Futai K. Regulation of Presynaptic Release Machinery by Cell Adhesion Molecules. Adv Neurobiol. 2023;33:333-356.

Kim J, Wulschner LEG, Oh WC, Ko J. Trans-synaptic mechanisms orchestrated by mammalian synaptic cell adhesion molecules. Bioessays. 2022 Nov;44(11):e2200134.

  1. Figure 1. Schematic depictions of the diverse modular structural organization of HSPGs

Comment: The resolution of Figure 1 needs to be improved. Figures have been re-done

  1. Table 1. The Biodiversity of HS-Proteoglycans.

Comment: Table 1 needs to be reformatted to improve the demarcation between the ‘Characteristics’ and ‘Distribution/Function’ columns. done

Authors response: Text has been amended.

  1. Comment: The manuscript should be revised to fix the uneven spacing between sentences throughout the manuscript.

Authors response: Extra spacing between sentences has been removed.

  1. Comment: Commas are missing at multiple places and the manuscript should be revised to resolve such minor grammatical errors.  

Authors response: The text has been reviewed, and commas have been added where appropriate.

Reviewer 2 Report

The review encapsulates the biology of the complex field of the superfamily of proteoglycans of multicellular organisms. There is considerable overlap with previous reviews of different authors, however the present review includes recently discovered HSPG family members in neuronal tissues, and in specific tissues relatively new PG isoforms that exhibit HS/CS glycosaminoglycan side chains. References are up to date and adequate. Conflict of interest is acknowledged.

Several comments are listed that if addressed would improve the flow of a mostly well-written review.

Line 47. should begin a new paragraph.

Line 47 "... 500 million year period...

Line 87 paragraph,  mention the general biological functions of serglycin.

Line 111 is unclear. The message seems to state that both GAG core protein interactions and the core protein cytoplasmic tail together are necessary to initiate cytoplasmic signal activation. Clarification is needed.

Line 141, describe the physiological roles of CD47-SIRP-alpha.

Line 143 paragraph, include examples of roles of CD44 isoforms substituted with different glycosaminoglycans.

Fig. 1 is overcrowded. More white space around the PG illustrations would help readers compare structural differences in the different PG.

Paragraphs in section 5 (HS Interactive proteins) should be re-edited. Compared to previous sections of the review the flow is lost in section 5. Information in lines 253-267 could be moved toward the introduction.

Line 270 could have "Perlecan" as a subsection heading, line 299, "Syndecan" as a subheading.

Line 357, List the high molecular weight range for HA. Line 360, is depolymerized HA fragmented HA?

Line 424, HSPG can help trigger downstream molecular switch activation (conventional kinase/phosphatase and G-protein GEF/GAP molecular switches). Does this review consider  HSPGs molecular switches?

English language is adaquate. Minor mistakes only (uppercase/lowercase etc...)

Author Response

Reviewer 2 comments

Major segments in the revised MS are highlighted, a marked up version of the revision is attached.

Top of Form

Comments and Suggestions for Authors

The review encapsulates the biology of the complex field of the superfamily of proteoglycans of multicellular organisms. There is considerable overlap with previous reviews of different authors, however the present review includes recently discovered HSPG family members in neuronal tissues, and in specific tissues relatively new PG isoforms that exhibit HS/CS glycosaminoglycan side chains. References are up to date and adequate. Conflict of interest is acknowledged.

Several comments are listed that if addressed would improve the flow of a mostly well-written review.

Line 47. should begin a new paragraph.

Authors response: Text has been amended.

Line 47 "... 500 million year period...

Authors response: Text has been amended.

Line 87 paragraph,  mention the general biological functions of serglycin.

Authors response: the content for 2.1 Cytoplasmic HS-PGs has been amended to include the biological functions of serglycin.

Serglycin is a small molecular weight (17.6 kDa core protein) intracellular heparin-proteoglycan present in cytotoxic T lymphocytes, leucocytes, and NK cells found in blood. Serglycin is also expressed in glioma, cross-talk of activated astrocytes with glioma cells enhances serglycin production and this is a predictive biomarker of poor survival [54, 55]. Serglycin can also be secreted and incorporated into the ECM [56, 57]. Serglycin displays highly divergent glycosylation patterns in different cell types. The Serglycin core protein has a central 16 amino acid region which is substituted with long GAG chains. Serglycin is expressed by cells of haematopoietic origin including neutrophils, lymphocytes, monocytes, macrophages, platelets, megakaryocytes, and mast cells [58-62] and by endothelial and embryonic stem cells [57, 63]. Intervertebral disc cells and chondrocytes also express Serglycin [64-66] with IL-1β or TNF-α increasing serglycin expression in-vitro. Serglycin levels are also elevated under inflammatory conditions during cartilage and IVD degeneration [64-66]. Heparin exclusively substitutes serglycin in connective tissue mast cells [67] however mucosal mast cells, activated monocytes and macrophages contain serglycin substituted with highly sulphated chondroitin-4,6-disulfate, quiescent monocytes contain serglycin substituted with chondroitin-4-sulfate [56].

Serglycin has important functional roles in the formation of storage granules which contain a range of bioactive molecules whose actions need to be controlled during storage [56]. In mast cells, these compounds include histamine, chymase, tryptase and carboxypeptidase. Elastase is stored in such granules in neutrophils, cytotoxic T cells store granzyme B, endothelial cells store tissue-type plasminogen activator while macrophages store TNF-α[68]. Serglycin has important roles in the retention of key inflammatory mediators in an inactive form inside storage granules and secretory vesicles which are released as active components when de-granulation occurs [56].   Release of these components at specific tissue sites can significantly influence the inflammatory process and innate immunity [69, 70].

Line 111 is unclear. The message seems to state that both GAG core protein interactions and the core protein cytoplasmic tail together are necessary to initiate cytoplasmic signal activation. Clarification is needed.

Authors response: The text has been amended to clarify the statement:

Cell signalling by syndecan can be is initiated in the cytoplasmic tails of the PG [82, 83], the core protein, or by GAG side chain interactions [79-81].

Line 141, describe the physiological roles of CD47-SIRP-alpha.

The following segment has been added to the CD47 section in the revised manuscript.

Author response

CD47 is a widely expressed integrin associated transmembrane protein encoded by the CD47 gene and is a member of the immunoglobulin superfamily that binds to integrins, thrombospondin-1 and signal-regulatory protein alpha (SIRP), a protein expressed by macrophages and dendritic cells [104]. Upon binding to CD47, SIRPα initiates a signaling cascade that results in the inhibition of phagocytosis [103]. Phosphorylation of the immunoreceptor tyrosine-based inhibition motifs present on the cytoplasmic tail of SIRPα initiates this response [105]. Subsequent binding and activation of SHP-1 and SHP-2 [src homology-2 (SH2)-domain containing protein tyrosine phosphatases] blocks phagocytosis, by preventing accumulation of myosin-IIA at the phagocytic synapse [106-109]. CD47 is a critical regulator of innate immune surveillance, the HS chain of CD47 is necessary for inhibition of T cell receptor signaling by thrombospondin-1 [110]. CD47 has important roles in the regulation of cellular migration, proliferation, and survival of vascular cells in innate and adaptive immune regulation [98, 105]. TSP-1 acts via CD47 to inhibit NO signaling in the vascular system and supports blood pressure regulation by limiting eNOS activation and endothelial-dependent vasorelaxation [106]. CD47 -SIRPα interactions represent a cell-cell communication system [102, 103, 111] termed an innate immune checkpoint sending a "don't eat me" signal to macrophages [108]. This allows tumor cells to avoid detection and be cleared by the immune system. Blockade of the CD47-SIRPα interaction using humanised antibodies to CD47 (Hu5F9-G4) has yielded promising results in a number of human malignancies including pediatric brain tumors: medulloblastoma, atypical teratoid rhabdoid tumors, primitive neuroectodermal tumor, pediatric glioblastoma, and diffuse intrinsic pontine glioma [107]. By targeting the CD47-SIRPα immunological checkpoint glioblastoma development can be inhibited and the activity of phagocytic, dendritic and T-lymphocytes enhanced promoting the efficiency of tumor cell clearance [109, 112-114].  

Line 143 paragraph, include examples of roles of CD44 isoforms substituted with different glycosaminoglycans.

Authors response: section 2.3.4 CD44 has been amended to the following, including roles of the different isoforms:

The major HA receptor CD44 exists as several widely distributed isoforms varying in size from 80-250 kDa. CD44s is the standard form, CD44E the epithelial form and CD44v variant isoforms.   CD44 can be substituted with all major classes of GAGs, HS sub-stituted CD44 has been termed epican [115] and is expressed by keratinocytes [116, 117].   Differential splicing of 11 variable exons in CD44 leads to 20 isoforms. CD44-HA inter-actions promote cell proliferation, migration, and invasion in inflammation and cancer progression [118]. Such interactions are complex and may promote or inhibit these pro-cesses.   CS and HS modified CD44 containing the v3 alternative exon encoding the consensus motif SGXG for GAG addition are found in CD44 isoforms v3-10 and v3,8-10 and occur in many tumors [119-121].   Epican is an HS modified form of CD44 [116].   Besides binding HA these GAG modified forms of CD44 can also bind growth factors impacting on cellular proliferation in inflammation and cancer progression.

The extracellular domain of CD44 can also bind collagens [117, 122] fibronectin, osteopontin and MMPs equipping CD44 with biosensory properties conveying cell transductive cues from the ECM.   CD44 has roles in cell-cell interactions, cell adhesion and migration, lymphocyte activation and homing, hematopoiesis, and tumour metas-tasis [123]. CD44 interactions with integrins, osteopontin, collagens, and MMPs have roles in the stabilisation and remodelling the CNS/PNS ECM [123, 124].   CD44 and integrins attached to neural cells generate focal adhesion complexes that generate traction forces and cell spreading and glioma cell invasion [125, 126]. Entrapment of HA by CD44 is important in the hydration and space-filling properties of the CNS/PNS ECM main-taining ionic gradients and hydration and compartmentalisation of neural tissues pre-serving neural niches and stem cell viability [127-130].   MMPs can lead to shedding of CD44 from the cell surface [131], soluble CD44 antagonises CD44 mediated cellular binding to HA and can modulate or inhibit tumour development [132] and tumor cell migration [133]. The transmembrane and cytoplasmic domains of CD44 have interactive properties with the cytoskeleton and roles in signal transduction which can up-regulate invasive tumor phenotypes and metastasis leading to matrix degradation, cancer and tumor cell migration [134].  

Fig. 1 is overcrowded. More white space around the PG illustrations would help readers compare structural differences in the different PG. Figure has been re-done.

Paragraphs in section 5 (HS Interactive proteins) should be re-edited. Compared to previous sections of the review the flow is lost in section 5. Information in lines 253-267 could be moved toward the introduction.

Line 270 could have "Perlecan" as a subsection heading, line 299, "Syndecan" as a subheading.

Authors response: Text amended to have subsections 5.1 Perlecan (line 399), and 5.2 Syndecan (line 430).

Line 357, List the high molecular weight range for HA. Line 360, is depolymerized HA fragmented HA? HMW HA is 1000kDa.

Line 424, HSPG can help trigger downstream molecular switch activation (conventional kinase/phosphatase and G-protein GEF/GAP molecular switches). Does this review consider  HSPGs molecular switches?

Author response

The following segments have been added to the revised manuscript.

2.3.1. Betaglycan

Betaglycan, a 250-280 kDa multifunctional transmembrane HS/CS co-receptor member of the TGF-beta superfamily [85, 86] forms a functional homodimer at the cell surface that binds inhibin, activin, FGF-2, Wnt and TGF [40, 87-94]. HS chains of betaglycan bind FGF-2, Wnt binds specific HS sequences independently of TGF-b binding [95]. Betaglycan, binds TGF-beta via its core protein thus betaglycan can bind several classes of growth factors through separate domains [96]. HS/CS dual modification of betaglycan modulates key cell signaling pathways including Wnt, TGF-β and FGF signaling [95]. Hyperactive Wnt/β-catenin signaling is linked to cancer progression and developmental abnormalities. Betaglycan controls Wnt3a bio-availability, independent of TGFβ co-receptor activity. HS and CS have opposing effects in TβRIII, HS inhibits Wnt signaling, whereas CS promotes Wnt3a signaling and is a key regulatory feature of betaglycan and may be considered a molecular switch over cellular behaviour during tissue development [97]. Betaglycan fragments released from the cell surface by plasmin and MMPs which act as sheddases is a circulating antagonist to the action of cell bound betaglycan.

2.4 Functional basis of HS/CS dual modification in PGs in control of cellular behaviour

2.4.1 HS/CS side chains and Syndecan core proteins promote Midkine and Pleiotrophin binding and tissue growth

CS chains of SDC-1 and 4 ectodomains are comprised of nonsulfated, 4-O-, 6-O-, and 4,6-O-disulfated N-acetylgalactosamine disaccharides. SDC-1 and 4 CS chains are significantly different, with a higher degree of sulfation evident in SDC-4 [135]. FGF2 binds to HS chains on SDC-1 and 4 as does midkine (MK) and pleiotrophin (PTN), MK and PTN also bind to CS chains. Stronger binding of MK and PTN has been observed in SDC-4 compared to SDC-1. Removal of CS chains decreases association and dissociation rate constants for MK, PTN, and bFGF in both SDCs, indicating that growth factor binding requires HS and CS chains. The core protein of SDC-1 also contributes to MK and PTN binding to growth factor and delivery to cell surface receptors [135]. HS and CS thus display supportive cooperative effects in growth factor binding in the SDCs and this effect is co-ordinated by interactions with their core proteins.

2.4.2 Co-ordinated actions of HS and CS in neural cell regulation

CS has neuronal growth-promoting and inhibitory properties which depend on CS sulfation, and cell and tissue context. HS however universally supports neuronal development [136, 137]. Unlike CS, where sulfation is an intracellular process, HS also undergoes extracellular modification by the 6-O-sulfatases Sulf1 and Sulf2 [138] which alters interactions with proteins and can promote cell signaling [139]. Two major classes of neuronal receptors bind HSPGs and CSPGs: type IIa receptor protein tyrosine phosphatases (RPTPσ, RPTPδ and LAR) and the Nogo receptors (NgR1 and NgR3) [140]. HSPG sulfation is crucial to the regulation of synaptic development and neurotransmission. This is achieved by bidirectional control of HS 6-O-sulfotransferase (hs6st) and Sulf1 activity [137]. CSPGs generally suppress neurite growth by blocking or interfering with neurite regulatory pathways, integrin, semaphorin, and EGFR pathways using intracellular processes involving calcium, RhoA/ROCK, Akt, PKC, and MAPK signaling to alter microtubule or actin cytoskeleton organization, gene expression and protein synthesis. RPTPσ and LAR bind to CSPGs with an affinity in the nanomolar range, binding is disrupted by pre-treatment with Chondroitinase ABC, demonstrating that this interaction is due to the CS-GAGs [141, 142]. In addition to binding HSPGs, RPTPσ and LAR are functional receptors for CSPGs and can bind neurocan and aggrecan. These interactions are sulfation-dependent, RPTPσ can bind CS-D, CS-E and DS, but not the more commonly found CS-A or CS-C [137, 143]. The CNS/PNS contains a diverse range of CSPGs with instructive roles in the development of embryonic neural axonal networks, and in responses displayed by neural cell populations in mature tissues to traumatic injury [144, 145]. Following brain trauma and spinal cord injury, a protective stabilizing CSPG-rich scar tissue is laid down at the defect site. Binding of HS and CS to RPTPσ and LAR mediate opposite effects on neural development and thus represent a molecular switch controlling neuronal behaviour [146].

Reviewer comment

HSPG can help trigger downstream molecular switch activation (conventional kinase/phosphatase and G-protein GEF/GAP molecular switches). Does this review consider  HSPGs molecular switches?

Author response

The major thrust of this review was to examine the cell regulatory properties of GAG side chains on proteoglycans. We thus limited our comments to the GAG level and did not delve into the intricacies of the regulatory properties of protein kinases in cell signalling pathways. We have added a few examples of the GAG regulatory properties which support the concept of GAGs acting as molecular switches.

HS acts as a Molecular switch over cellular behavior in specific tissue contexts.

HS and CS-PGs proteoglycans regulate numerous cell surface signaling events, typically producing opposing effects on cell regulation. CSPGs inhibit nerve regeneration through RPTPσ. When decorated with CS RPTPσ inhibits sensory neuron extension, however HS substitution on RPTPσ promotes neuronal growth. Crystallographic analyses of HSPG-CSPG binding sites on RPTPσ reveals these can accommodate diverse GAGs with comparable affinities. HS induces RPTPσ ectodomain oligomerization in solution, which is inhibited by CS. RPTPσ and HSPGs share a punctate colocalization in sensory neurons in-vitro, contrasting with CSPGs which are distributed throughout the ECM. This has led to a proposal that HS and CS-PGs can exert opposing effects on neuronal extension by controlling the oligomerization of RPTPσ [1]. A synthetic HS polymer has subsequently shown that the corneal epithelium and innervation can be regenerated by this polymer, restoring corneal transparency and downregulating myofibroblasts activity and scarring after experimental injury of these tissues [2]. Perlecan can either prevent or promote the progression of cancers and metastatic disease. Breast, prostate, lung, and renal cancers all preferentially metastasize to bone in a dense, perlecan-rich environment. However perlecan can also promote production of MMPs, SULFs, and heparanase, in the tumor stroma which modify the actions of perlecan detrimentally effecting cell adhesion, invasion, angiogenesis and tumor development. Thus perlecan can be considered a molecular switch in tumor development based on modification of its HS and other functional core protein components [3-5].

HS has guidance roles in assembly of neural networks.

Roundabout 1, or Robo1, is a cell surface signaling molecule important in axonal guidance during neural network formation. Interaction of Robo-1 with HS and members of the Slit protein family confers its axonal guidance activity [6]. O-sulfate groups on HS chains have critical roles in interactions with Slit [7]. Robo-1 interacts with the HS tetrasaccharide IdoA-GlcNS6S-IdoA2S-GlcNS6S-(CH2)5NH2 [6]. Robo-Slit interactions are modulated by HS, the HS chains of GP1 promote dimerisation of Slit and formation of a complex with Robo-1 aiding in axonal guidance in neural network formation [7-9].

  1. Coles CH, Shen Y, Tenney AP, Siebold C, Sutton GC, Lu W, Gallagher JT, Jones EY, Flanagan JG, Aricescu AR. Proteoglycan-specific molecular switch for RPTPσ clustering and neuronal extension. Science. 2011 Apr 22;332(6028):484-8.
  2. Alcalde I, Sánchez-Fernández C, Del Olmo-Aguado S, Martín C, Olmiere C, Artime E, Quirós LM, Merayo-Lloves J. Synthetic Heparan Sulfate Mimetic Polymer Enhances Corneal Nerve Regeneration and Wound Healing after Experimental Laser Ablation Injury in Mice. Polymers (Basel). 2022 Nov 15;14(22):4921.
  3. Cruz LA, Tellman TV, Farach-Carson MC. Flipping the Molecular Switch: Influence of Perlecan and Its Modifiers in the Tumor Microenvironment. Adv Exp Med Biol. 2020;1245:133-146.
  4. Elgundi Z, Papanicolaou M, Major G, Cox TR, Melrose J, Whitelock JM, Farrugia BL. Cancer Metastasis: The Role of the Extracellular Matrix and the Heparan Sulfate Proteoglycan Perlecan. Front Oncol. 2020 Jan 17;9:1482.
  5. Steeg PS. Tumor metastasis: mechanistic insights and clinical challenges. Nat Med. (2006) 12:895–904.
  6. Gao Q, Chen CY, Zong C, Wang S, Ramiah A, Prabhakar P, Morris LC, Boons GJ, Moremen KW, Prestegard JH. Structural Aspects of Heparan Sulfate Binding to Robo1-Ig1-2. ACS Chem Biol. 2016 Nov 18;11(11):3106-3113.
  7. Ronca F, Andersen JS, Paech V, Margolis RU. Characterization of Slit protein interactions with glypican-1. J Biol Chem. 2001 Aug 3;276(31):29141-7.
  8. Zhang F, Ronca F, Linhardt RJ, Margolis RU. Structural determinants of heparan sulfate interactions with Slit proteins. Biochem Biophys Res Commun. 2004 Apr 30;317(2):352-7.
  9. Liang Y, Annan RS, Carr SA, Popp S, Mevissen M, Margolis RK, Margolis RU. Mammalian homologues of the Drosophila slit protein are ligands of the heparan sulfate proteoglycan glypican-1 in brain. J Biol Chem. 1999 Jun 18;274(25):17885-92.
